# BAYESIAN ONLINE META-LEARNING

## ABSTRACT

Neural networks are known to suffer from catastrophic forgetting when trained on sequential datasets. While there have been numerous attempts to solve this problem for large-scale supervised classification, little has been done to overcome catastrophic forgetting for few-shot classification problems. Few-shot meta-learning algorithms often require all few-shot tasks to be readily available in a batch for training. The popular gradient-based model-agnostic meta-learning algorithm (MAML) is a typical algorithm that suffers from these limitations. This work introduces a Bayesian online meta-learning framework to tackle the catastrophic forgetting and the sequential few-shot tasks problems. Our framework incorporates MAML into a Bayesian online learning algorithm with Laplace approximation or variational inference. This framework enables few-shot classification on a range of sequentially arriving datasets with a single meta-learned model and training on sequentially arriving few-shot tasks. The experimental evaluations demonstrate that our framework can effectively prevent catastrophic forgetting and is capable of online meta-learning in various few-shot classification settings.

## 1 INTRODUCTION

Image classification models and algorithms often require an enormous amount of labelled examples for training to achieve state-of-the-art performance. Labelled examples can be expensive and time-consuming to acquire. Human visual systems, on the other hand, are able to recognise new classes after being shown a few labelled examples. Few-shot classification (Miller et al., 2000; Li et al., 2004; 2006; Lake et al., 2011) tackles this issue by learning to adapt to unseen classes (known as **novel classes**) with very few labelled examples from each class. Recent works show that meta-learning provides promising approaches to few-shot classification problems (Santoro et al., 2016; Finn et al., 2017; Li et al., 2017; Ravi & Larochelle, 2017). Meta-learning or learning-to-learn (Schmidhuber, 1987; Thrun & Pratt, 1998) takes the learning process a level deeper – instead of learning from the labelled examples in the training classes (known as **base classes**), meta-learning learns the example-learning process. The training process in meta-learning that utilises the base classes is called the **meta-training** stage, and the evaluation process that reports the few-shot performance on the novel classes is known as the **meta-evaluation** stage.

Despite being a promising solution to few-shot classification problems, meta-learning methods suffer from several limitations:

1. **Unable to continually learn from sequential few-shot tasks:** It is mandatory to have all base classes readily available for meta-training. Such meta-learning algorithms often require sampling a number of few-shot tasks in every iteration for optimisation.

2. **Unable to retain few-shot classification ability on sequential datasets that have evident distributional shift:** A meta-learned model is restricted to perform few-shot classification on a specific dataset, in the sense that the base and novel classes have to originate from the same dataset distribution. A meta-learned model loses its few-shot classification ability on previous datasets as new ones arrive subsequently for meta-training.

We emphasise that the task mentioned in this paper refers to the **few-shot task** for meta-learning. This paper considers meta-learning a single model for few-shot classification in *the sequential datasets and sequential few-shot tasks settings* respectively.

We introduce a Bayesian online meta-learning framework that can train a few-shot learning model under the sequential few-shot tasks setting and train a model that is applicable to a broader scope of few-shot classification datasets by overcoming catastrophic forgetting. We extend the Bayesian online learning (BOL) framework (Opper, 1998) to a Bayesian online meta-learning framework using the model-agnostic meta-learning (MAML) algorithm (Finn et al., 2017). MAML finds a good model parameter initialisation (called **meta-parameters**) that can quickly adapt to novel classes using very few labelled examples, while BOL provides a principled framework for finding the posterior of the model parameters. Our framework aims to combine both BOL and MAML to find the posterior of the meta-parameters. Our work builds on Ritter et al. (2018a) which combines the BOL framework and Laplace approximation with block-diagonal Kronecker-factored Fisher approximation, and Nguyen et al. (2018) which uses variational inference with BOL to overcome catastrophic forgetting in large-scale supervised classification.

An important reason to implement Bayesian inference over non-Bayesian methods for an online setting is that BOL provides a grounded framework that suggests using the previous posterior as the prior recursively. Bayesian inference inherits an advantage for robust meta-learning (Yoon et al., 2018) to overcome training instability problems addressed by Antoniou et al. (2019). BOL implicitly keeps a memory on previous knowledge via the posterior, in contrast to recent online meta-learning methods that explicitly accumulate previous data in a task buffer (Finn et al., 2019; Zhuang et al., 2019). Explicitly keeping a memory on previous data often triggers an important question: *how should the carried-forward data be processed in future task rounds, in order to accumulate knowledge?* Finn et al. (2019) update the meta-parameters at each iteration using previous few-shot tasks in the task buffer. This defeats the purpose of online learning, which by definition means to update the parameters each round using only the new data encountered. Having to re-train on previous data to avoid forgetting also increases the training time as the data accumulate (Finn et al., 2019; He et al., 2019). Certainly one can clamp the amount of data at some maximal limit and sample from the buffer, but the final performance of such an algorithm would be dependent on the samples being informative and of good quality which may vary across different seed runs. In contrast to memorising the datasets, having an implicit memory via the posterior automatically deals with the question on how to process carried-forward data and allows a better carry forward in previous experiences. @@@ New

Below are the contributions we make in this paper:

- We develop the Bayesian online meta-learning (BOML) framework for sequential few-shot classification problems. Under this framework we introduce the algorithms Bayesian online meta-learning with Laplace approximation (BOMLA) and Bayesian online meta-learning with variational inference (BOMVI).
- We propose a simple approximation to the Fisher corresponding to the BOMLA algorithm that carries over the desirable block-diagonal Kronecker-factored structure from the Fisher approximation in the non-meta-learning setting.
- We demonstrate that BOML can overcome catastrophic forgetting in the sequential few-shot datasets setting with apparent distributional shift in the datasets.
- We demonstrate that BOML can continually learn to few-shot classify the novel classes in the sequential meta-training few-shot tasks setting.

## 2 META-LEARNING

Most meta-learning algorithms comprise an inner loop for example-learning and an outer loop that learns the example-learning process. Such algorithms often require sampling a meta-batch of tasks at each iteration, where a **task** is formed by sampling a subset of classes from the pool of base classes or novel classes during meta-training or meta-evaluation respectively. The $N$-way $K$-shot task, for instance, refers to sampling $N$ classes and using $K$ examples per class for few-shot quick adaptation.

An offline meta-learning algorithm learns a few-shot classification model only for a specific dataset $\mathcal{D}_{t+1}$ where all base classes of $\mathcal{D}_{t+1}$ have to be readily available for meta-training. For notational convenience, we drop the $t+1$ subscript in this section, as there is only one dataset involved in offline meta-learning. The dataset $\mathcal{D}_{t+1}$ is divided into the set of base classes $\widetilde{\mathcal{D}}$ and novel classes $\widehat{\mathcal{D}}$ for meta-training and meta-evaluation respectively. Upon completing meta-training on the base class

set $\widetilde{\mathcal{D}}$, the goal of few-shot classification is to perform well on an unseen task $\widehat{\mathcal{D}}^*$ sampled from the novel class set $\widehat{\mathcal{D}}$ after a quick adaptation on a small subset $\widehat{\mathcal{D}}^{*,S}$ (known as the **support set**) of $\widehat{\mathcal{D}}^*$. The performance of this unseen task is evaluated on the **query set** $\widehat{\mathcal{D}}^{*,Q}$, where $\widehat{\mathcal{D}}^{*,Q} = \widehat{\mathcal{D}}^* \backslash \widehat{\mathcal{D}}^{*,S}$. Since $\widehat{\mathcal{D}}$ is not accessible during meta-training, this support-query split is mimicked on the base class set $\widetilde{\mathcal{D}}$ for meta-training.

**Model-agnostic meta-learning**  We are interested in the well-known meta-learning algorithm MAML (Finn et al., 2017). Each updating step of MAML aims to improve the ability of the meta-parameters to act as a good model initialisation for a quick adaptation on unseen tasks. Each iteration of the MAML algorithm samples $M$ tasks from the base class set $\widetilde{\mathcal{D}}$ and runs a few steps of stochastic gradient descent (SGD) for an inner loop task-specific learning. The number of tasks sampled per iteration is known as the **meta-batch size**. For task $m$, the inner loop outputs the task-specific parameters $\tilde{\theta}^m$ from a $k$-step SGD quick adaptation on the objective $\mathcal{L}(\theta, \widetilde{\mathcal{D}}^{m,S})$ with the support set $\widetilde{\mathcal{D}}^{m,S}$ and initialised at $\theta$:

$$\tilde{\theta}^m = SGD_k(\mathcal{L}(\theta, \widetilde{\mathcal{D}}^{m,S})), \tag{1}$$

where $m = 1, \ldots, M$. The outer loop gathers all task-specific adaptations to update the meta-parameters $\theta$ using the loss $\mathcal{L}(\tilde{\theta}^m, \widetilde{\mathcal{D}}^{m,Q})$ on the query set $\widetilde{\mathcal{D}}^{m,Q}$.

The overall MAML optimisation objective is

$$\arg\min_\theta \frac{1}{M} \sum_{m=1}^M \mathcal{L}(SGD_k(\mathcal{L}(\theta, \widetilde{\mathcal{D}}^{m,S})), \widetilde{\mathcal{D}}^{m,Q}). \tag{2}$$

Like most offline meta-learning algorithms, MAML requires all base classes to be readily available for tasks-sampling at each iteration. We aim to overcome this limitation by meta-learning a model that can few-shot classify unseen tasks from the novel classes, while the tasks from the base classes arrive sequentially for meta-training. MAML also assumes a stationary task distribution during meta-training and meta-evaluation. Under this assumption, a meta-learned model is only applicable to a specific dataset distribution. When the model encounters a sequence of datasets with apparent distributional shift, it loses the few-shot classification ability on previous datasets as new ones arrive for meta-training. Our work also aims to meta-learn a single model for few-shot classification on multiple datasets that arrive sequentially for meta-training. We achieve these two goals by incorporating MAML into the BOL framework to give the *Bayesian online meta-learning* (BOML) framework that finds the posterior of the meta-parameters.

## 3 OVERVIEW OF OUR BAYESIAN ONLINE META-LEARNING APPROACH

Our central contribution is to extend the benefits of meta-learning to the Bayesian online scenario, thereby training models that can generalise across tasks whilst dealing with parameter uncertainty in the setting of sequential tasks or sequential datasets.

**Sequential datasets setting**  In this setting, online meta-training occurs sequentially on the datasets $\mathcal{D}_1, \ldots, \mathcal{D}_T$. Each dataset $\mathcal{D}_i$ can be seen as a knowledge domain with an associated underlying task distribution $p(\mathcal{T}_i)$. A newly-arrived $\mathcal{D}_{t+1}$ is separated into the base class set $\widetilde{\mathcal{D}}_{t+1}$ and novel class set $\widehat{\mathcal{D}}_{t+1}$ for meta-training and meta-evaluation respectively, where the tasks in these two stages are drawn from the task distribution $p(\mathcal{T}_{t+1})$.

**Sequential tasks setting**  The sequential tasks setting only involves one dataset $\mathcal{D}$ with an associated underlying task distribution $p(\mathcal{T})$, where $\mathcal{D}$ is separated into the base and novel class sets. In this setting, $\widetilde{\mathcal{D}}_1, \ldots, \widetilde{\mathcal{D}}_{t+1}$ denote the non-overlapping tasks formed from the base class set and they arrive sequentially for meta-training. These tasks $\widetilde{\mathcal{D}}_1, \ldots, \widetilde{\mathcal{D}}_{t+1}$ and the meta-evaluation tasks are drawn from the task distribution $p(\mathcal{T})$.

Notationally, for both sequential tasks and sequential datasets settings, let $\widetilde{\mathcal{D}}_{t+1}^S$ and $\widetilde{\mathcal{D}}_{t+1}^Q$ denote the collection of support sets and query sets respectively from $\widetilde{\mathcal{D}}_{t+1}$, so that $\widetilde{\mathcal{D}}_{t+1} = \widetilde{\mathcal{D}}_{t+1}^S \cup \widetilde{\mathcal{D}}_{t+1}^Q$.

We are interested in a MAP estimate $\theta^* = \arg\max_\theta p(\theta|\widetilde{\mathcal{D}}_{1:t+1})$. Using Bayes' rule on the posterior gives the recursive formula

$$p(\theta|\widetilde{\mathcal{D}}_{1:t+1}) \propto p(\widetilde{\mathcal{D}}_{t+1}^S, \widetilde{\mathcal{D}}_{t+1}^Q|\theta)\, p(\theta|\widetilde{\mathcal{D}}_{1:t}) \tag{3}$$

$$= p(\widetilde{\mathcal{D}}_{t+1}^Q|\theta, \widetilde{\mathcal{D}}_{t+1}^S)\, p(\widetilde{\mathcal{D}}_{t+1}^S|\theta)\, p(\theta|\widetilde{\mathcal{D}}_{1:t}) \tag{4}$$

$$= \left\{ \int p(\widetilde{\mathcal{D}}_{t+1}^Q|\tilde{\theta})\, p(\tilde{\theta}|\theta, \widetilde{\mathcal{D}}_{t+1}^S)\, d\tilde{\theta} \right\} p(\widetilde{\mathcal{D}}_{t+1}^S|\theta)\, p(\theta|\widetilde{\mathcal{D}}_{1:t}) \tag{5}$$

where Eq. (3) follows from the assumption that each dataset is independent given $\theta$.

From the meta-learning perspective, the parameters $\tilde{\theta}$ introduced in Eq. (5) can be viewed as the task-specific parameters in MAML. There are various choices for the distribution $p(\tilde{\theta}|\theta, \widetilde{\mathcal{D}}_{t+1}^S)$ in Eq. (5). In particular if we choose to set it as the deterministic function of taking several steps of SGD on loss $\mathcal{L}$ with the support set collection $\widetilde{\mathcal{D}}_{t+1}^S$ and initialised at $\theta$, we have

$$p(\tilde{\theta}|\theta, \widetilde{\mathcal{D}}_{t+1}^S) = \mathbb{1}\{\tilde{\theta} = SGD_k(\mathcal{L}(\theta, \widetilde{\mathcal{D}}_{t+1}^S))\}. \tag{6}$$

and this recovers the MAML inner loop with SGD quick adaptation in Eq. (1). The recursion given by Eq. (5) forms the basis of our approach and the remainder of this paper explains how we implement this. In order to do so we give a mini tutorial in Appendix A on Bayesian online learning, Laplace approximation and variational continual learning.

## 4 BAYESIAN ONLINE META-LEARNING IMPLEMENTATION

This section demonstrates how we arrive at the algorithms Bayesian online meta-learning with Laplace approximation (BOMLA) and Bayesian online meta-learning with variational inference (BOMVI) by implementing Laplace approximation and variational continual learning respectively to the posterior of the BOML framework in Eq. (5). These algorithms from the grounded BOML framework are useful for online training on the sequential few-shot classification datasets or tasks.

### 4.1 BAYESIAN ONLINE META-LEARNING WITH LAPLACE APPROXIMATION

We discover that the Laplace approximation method provides a well-fitted meta-training framework for Bayesian online meta-learning in Eq. (5). Each updating step in the approximation procedure can be modified to correspond to the meta-parameters for few-shot classification, instead of the model parameters for large-scale supervised classification.

Laplace approximation rationalises the use of a Gaussian approximate posterior by Taylor expanding the log-posterior around a mode up to the second order, as described in Appendix A.2. The second order term corresponds to the log-probability of a Gaussian distribution. The BOML framework in Section 3 with a Gaussian approximate posterior $q$ of mean and precision $\phi_t = \{\mu_t, \Lambda_t\}$ from the Laplace approximation gives a MAP estimate:

$$\theta^* = \arg\max_\theta \left\{ \log \int p(\widetilde{\mathcal{D}}_{t+1}^Q|\tilde{\theta}) p(\tilde{\theta}|\theta, \widetilde{\mathcal{D}}_{t+1}^S)\, d\tilde{\theta} + \log p(\widetilde{\mathcal{D}}_{t+1}^S|\theta) - \frac{1}{2}(\theta - \mu_t)^T \Lambda_t (\theta - \mu_t) \right\}. \tag{7}$$

For an efficient optimisation, we use the deterministic $\tilde{\theta}$ in Eq. (6). The objective in Eq. (7) can be *batched* (for sequential tasks) or *meta-batched* (for sequential datasets). This leads to minimising the objective

$$f_{t+1}^{\text{BOMLA}}(\theta, \mu_t, \Lambda_t) = -\frac{1}{M}\sum_{m=1}^{M} \log p(\widetilde{\mathcal{D}}_{t+1}^{m,Q}|\tilde{\theta}^m) - \frac{1}{M}\sum_{m=1}^{M} \log p(\widetilde{\mathcal{D}}_{t+1}^{m,S}|\theta) + \frac{1}{2}(\theta - \mu_t)^T \Lambda_t (\theta - \mu_t), \tag{8}$$

where $\tilde{\theta}^m = SGD_k(\mathcal{L}(\theta, \widetilde{\mathcal{D}}_{t+1}^{m,S}))$ for $m = 1, \ldots, M$. In the sequential datasets setting $M$ denotes the number of tasks sampled per iteration, whereas in the sequential tasks setting $M$ denotes the number of batches per epoch. The first term of the objective in Eq. (8) corresponds to the MAML objective in Eq. (2) with a cross-entropy loss, the second term can be viewed as the pre-adaptation loss on the support set and the last term can be seen as a regulariser.

### 4.2 HESSIAN APPROXIMATION

We calculate a block-diagonal Kronecker-factored Hessian approximation in order to update the precision $\Lambda_t$, as explained in Appendix A.3. The Hessian approximations in both sequential datasets and sequential tasks settings are very similar, except that the sequential datasets setting averages over the meta-batch size and the sequential tasks setting averages over the number of batches.

The Hessian matrix corresponding to the first term of the BOMLA objective in Eq. (8) is

$$\widetilde{H}_{t+1}^{ij} = \frac{1}{M} \sum_{m=1}^{M} -\frac{\partial^2}{\partial \theta^{(i)} \partial \theta^{(j)}} \log p(\widetilde{\mathcal{D}}_{t+1}^{m,Q} | \tilde{\theta}^m)) \bigg|_{\theta = \mu_{t+1}}. \tag{9}$$

It is worth noting that the BOMLA Hessian deviates from the original BOL Hessian in Appendix A.2. This requires deriving an adjusted approximation to the Hessian with some further assumptions.

The BOL Hessian for a single data point can be approximated using the Fisher information matrix $F$ to ensure its positive semi-definiteness (Martens & Grosse, 2015):

$$F = \mathbb{E}_{x,y} \left[ \frac{d}{d\theta} \log p(y|x,\theta) \frac{d}{d\theta} \log p(y|x,\theta)^T \right]. \tag{10}$$

Each $(x,y)$ pair for the Fisher in BOMLA is associated to a task (or a batch) $m$. The Fisher information matrix $\widetilde{F}$ corresponding to the BOMLA Hessian in Eq. (9) for a single data point is

$$\widetilde{F} = \frac{1}{M} \sum_{m=1}^{M} \mathbb{E}_{x,y} \left[ \left( \frac{\partial \tilde{\theta}^m}{\partial \theta} \right) \frac{d}{d\tilde{\theta}^m} \log p(y|x,\tilde{\theta}^m) \frac{d}{d\tilde{\theta}^m} \log p(y|x,\tilde{\theta}^m)^T \left( \frac{\partial \tilde{\theta}^m}{\partial \theta} \right)^T \right]. \tag{11}$$

The additional Jacobian matrix $\frac{\partial \tilde{\theta}^m}{\partial \theta}$ breaks the Kronecker-factored structure described by Martens & Grosse (2015) for the original Fisher in Eq. (10).

The results in Finn et al. (2017) show that the first step of the quick adaptation in $\tilde{\theta}^m$ contributes the largest change to the meta-evaluation objective, and the remaining adaptation steps give a relatively small change to the objective. It is reasonable to assume that the quick adaptation is a one-step SGD for Fisher approximation:

$$\tilde{\theta}^m = \theta - \nabla_\theta \mathcal{L}(\theta, \widetilde{\mathcal{D}}_{t+1}^{m,S}). \tag{12}$$

By imposing this assumption, the $(i,j)$-th entry of the Jacobian term can be interpreted as

$$\left( \frac{\partial \tilde{\theta}^m}{\partial \theta} \right)^{ij} = I^{ij} - \frac{\partial^2 (-\log p(\widetilde{\mathcal{D}}_{t+1}^{m,S} | \theta))}{\partial \theta^{(i)} \partial \theta^{(j)}}, \tag{13}$$

where $I$ is the corresponding identity matrix and the objective $\mathcal{L}$ involved is the negative log-likelihood. The Hessian for a single data point in the second term of Eq. (13) can be approximated by $F$ in Eq. (10) via the usual block-diagonal Kronecker-factored approximation. Putting the Jacobian back into Eq. (11) and expanding the factors give terms that multiply two or more Kronecker products together. The detailed derivation of $\widetilde{F}$ is explained in Appendix A.3.1. We introduce the posterior regulariser $\lambda$ when updating the precision: $\Lambda_{t+1} = \lambda \widetilde{H}_{t+1} + \Lambda_t$ and the rationale for introducing $\lambda$ is explained in Appendix A.3.2. The pseudo-code of the BOMLA algorithm can be found in Appendix B.1.

### 4.3 BAYESIAN ONLINE META-LEARNING WITH VARIATIONAL INFERENCE

The variational continual learning (VCL) framework (Nguyen et al., 2018) is directly applicable to BOML. This section demonstrates how we arrive at the BOMVI algorithm by implementing VCL to the posterior of the BOML framework in Eq. (5).

As described in Appendix A.4, VCL approximates the posterior by minimising the KL-divergence over some pre-determined approximate posterior family $\mathcal{Q}$. Fitting the BOML posterior in Eq. (5) into the VCL framework gives the approximate posterior:

$$q(\theta|\phi_{t+1}) = \underset{q \in \mathcal{Q}}{\arg\min} \, D_{\mathrm{KL}} \left( q(\theta|\phi) \middle\| \left\{ \int p(\widetilde{\mathcal{D}}_{t+1}^Q | \tilde{\theta}) \, p(\tilde{\theta}|\theta, \widetilde{\mathcal{D}}_{t+1}^S) \, d\tilde{\theta} \right\} p(\widetilde{\mathcal{D}}_{t+1}^S | \theta) \, q(\theta|\phi_t) \right). \tag{14}$$

Similar to BOMLA, we use the deterministic $\tilde{\theta}$ in Eq. (6), and the objective in Eq. (14) can be *batched* (for sequential tasks) or *meta-batched* (for sequential datasets). This leads to minimising the objective

$$f_{t+1}^{\mathrm{BomVI}}(\phi, \phi_t) = -\frac{1}{M} \sum_{m=1}^{M} \mathbb{E}_{q(\theta|\phi)} \big[ \log p(\widetilde{\mathcal{D}}_{t+1}^{m,Q}|\tilde{\theta}^m) \big] - \frac{1}{M} \sum_{m=1}^{M} \mathbb{E}_{q(\theta|\phi)} \big[ \log p(\widetilde{\mathcal{D}}_{t+1}^{m,S}|\theta) \big]$$
$$+ D_{\mathrm{KL}}(q(\theta|\phi)\|q(\theta|\phi_t)), \tag{15}$$

where $\tilde{\theta}^m = SGD_k(\mathcal{L}(\theta, \widetilde{\mathcal{D}}_{t+1}^{m,S}))$ for $m = 1, \dots, M$. In the sequential datasets setting $M$ denotes the number of tasks sampled per iteration, whereas in the sequential tasks setting $M$ denotes the number of batches per epoch. We use a Gaussian mean-field approximate posterior $q(\theta|\phi_t) = \prod_{d=1}^{D} N(\mu_{t,d}, \sigma_{t,d}^2)$, where $\phi_t = \{\mu_{t,d}, \sigma_{t,d}\}_{d=1}^{D}$, $D = \dim(\theta)$ and the objective in Eq. (15) is minimised over $\phi$. The pseudo-code of the BOMVI algorithm can be found in Appendix B.1.

The first term in Eq. (15) is rather cumbersome to estimate in optimisation. To compute its Monte Carlo estimator, we have to generate samples $\theta_r \sim q$ for $r = 1, \dots, R$, and run a quick adaptation on each sampled meta-parameters $\theta_r$ before evaluating its log-likelihood. This is computationally intensive and it gives an estimator with large variance. We propose a workaround by modifying the inner loop SGD quick adaptation, and the details can be found in Appendix B.2.

## 5 RELATED WORK

**Online Meta-Learning**  There are two common problem settings in the current online meta-learning works:

- **Underlying task distribution:** Sequential tasks are assumed to originate from the same underlying task distribution $p(\mathcal{T})$ in this setting. Our work in the sequential tasks setting belongs to this category. Denevi et al. (2019) introduce the online-within-online (OWO) and online-within-batch (OWB) settings, where OWO encounters tasks and examples within tasks sequentially while OWB encounters tasks sequentially but examples within tasks are in batch. The BOML framework in the sequential tasks setting corresponds to the OWB setting. On the other hand, our work in the sequential datasets setting is novel in overcoming few-shot catastrophic forgetting, where the goal is to few-shot classify tasks drawn from a sequence of distributions $p(\mathcal{T}_1), \dots, p(\mathcal{T}_T)$ as explained in Section 3. He et al. (2019), Harrison et al. (2019) and Jerfel et al. (2019) look into continual meta-learning for non-stationary task distributions where the task boundaries are unknown to the model. Jerfel et al. (2019) consider a latent task structure to adapt to the non-stationary task distributions.

- **Regret minimisation:** In this setting, the goal is to minimise the regret function, and the assumptions are made on the loss function rather than the task distribution. Recent works Finn et al. (2019); Zhuang et al. (2019) belong to this category, where the aim is to compete with the best meta-learner and supersede it. These methods accumulate data as they arrive and meta-learn using all data acquired so far. Data accumulation is not desirable as the algorithmic complexity of training grows with the amount of data accumulated, and training time increases as new data arrive (Finn et al., 2019; He et al., 2019). The agent will eventually run out of memory for a long sequence of data. The BOML framework on the other hand is advantageous, as it only takes the posterior of the meta-parameters into consideration during optimisation. This gives a framework with an algorithmic complexity independent of the length of the dataset sequence.

**Offline Meta-Learning**  Previous meta-learning works attempt to solve few-shot classification problems in an offline setting, under the assumption of having a stationary task distribution during meta-training and meta-evaluation. A single meta-learned model is aimed to few-shot classify one specific dataset with all base classes of the dataset readily available in a batch for meta-training. There are two general frameworks for the offline meta-learning setting:

- **Probabilistic framework:** The MAML algorithm can be cast into a probabilistic inference problem (Finn et al., 2018) or with a hierarchical Bayesian structure (Grant et al.,

2018; Yoon et al., 2018). Grant et al. (2018) discuss the use of a Laplace approximation in the task-specific inner loop to improve MAML using the curvature information, whilst Yoon et al. (2018) use Stein Variational Gradient Descent (SVGD) for task-specific learning. Gordon et al. (2019) implement probabilistic inference by considering the posterior predictive distribution with amortised networks.

- **Non-probabilistic framework:** Gradient-based meta-learning (Finn et al., 2017; Nichol et al., 2018; Rusu et al., 2019) updates the meta-parameters by accumulating the gradients of a meta-batch of task-specific inner loop updates. The meta-parameters will be used as a model initialisation for a quick adaptation on the novel classes. Metric-based meta-learning (Koch et al., 2015; Vinyals et al., 2016; Snell et al., 2017) utilises the metric distance between labelled examples. Such methods assume that base and novel classes are from the same dataset distribution, and the metric distance estimations can be generalised to the novel classes upon meta-learning the base classes.

**Continual Learning**   Modern continual learning works (Goodfellow et al., 2013; Lee et al., 2017; Zenke et al., 2017) focus primarily on large-scale supervised learning, in contrast to our work that looks into continual few-shot classification across sequential tasks and datasets. Wen et al. (2018) utilise few-shot learning to improve on overcoming catastrophic forgetting via logit matching on a small sample from the previous tasks. The online learning element in this paper is closely related to (Kirkpatrick et al., 2017; Zenke et al., 2017; Ritter et al., 2018a; Nguyen et al., 2018) that overcome catastrophic forgetting for large-scale supervised classification. In particular, our work builds on the online Laplace approximation method in (Ritter et al., 2018a). We extend this to the meta-learning scenario to avoid forgetting in few-shot classification problems. Nguyen et al. (2018) provide the alternative of using variational inference instead of Laplace approximation for approximating the posterior. It is a reasonable approach to adapt variational approximation methods to approximate the posterior of the meta-parameters by adjusting the KL-divergence objective.

## 6 EXPERIMENTS

### 6.1 OMNIGLOT: SEQUENTIAL TASKS

We run the sequential tasks experiment on the Omniglot dataset. To increase the difficulty level, we split the datasets based on the alphabets (super-classes) instead of the characters (classes). The goal of this experiment is to classify the 5-way 5-shot novel tasks sampled from the meta-evaluation alphabets. The experimental details and the alphabet splits can be found in Appendix C.1.

We compare our algorithms to the following baselines:

1. **Train-On-Everything (TOE):**  When a new task (or dataset) arrives for meta-training, we randomly re-initialise the meta-parameters and perform *meta-training on all tasks (or datasets) encountered so far*. Once meta-training is completed in this stage, we do not update the posterior of the meta-parameters like we would in BOMLA and BOMVI.

2. **Train-From-Scratch (TFS):**  Upon the arrival of a new task (or dataset), we randomly re-initialise the meta-parameters and *meta-train only on the newly-arrived task (or dataset)*. Similar to TOE, the posterior of the meta-parameters is not updated in TFS.

3. **Follow The Meta-Leader (FTML):**  We introduce a slight modification to FTML (Finn et al., 2019) on its evaluation method, as FTML is not designed for few-shot learning on unseen tasks. In our experiment, we apply `Update-Procedure` in FTML to the data from unseen tasks, rather than the data from the same training task as in the original FTML.

@@@ New2: FTML & error-band

As the tasks arrive sequentially for meta-training, Figure 1 shows that BOMLA and BOMVI can accumulate the few-shot classification ability on the novel tasks over time. The knowledge acquired from previous meta-training tasks are carried forward in the form of a posterior, which is then used as the prior when a new task arrives for meta-training. The baselines TOE and TFS have similar performances. Despite having access to all previous tasks, TOE shows no positive forward transfer in the meta-evaluation accuracy each time it encounters a new task. BOMLA with $\lambda = 0.1$ gives the best performance in this experiment.

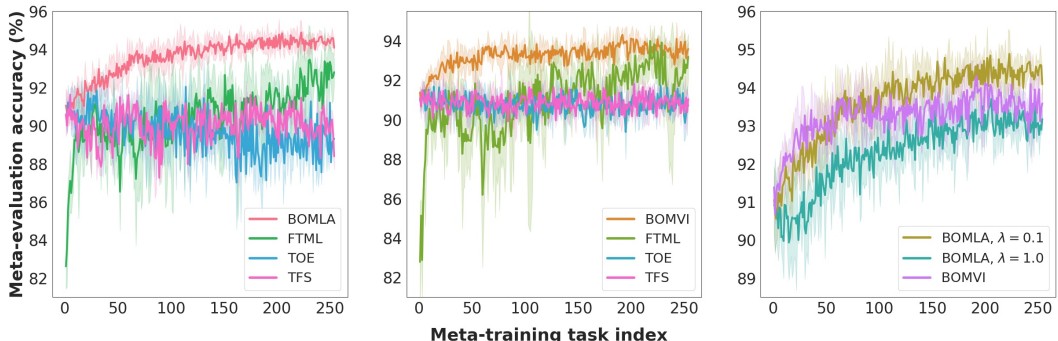

Figure 1: Meta-evaluation accuracy across 3 seed runs on the novel tasks along meta-training. **Left**: compares BOMLA to the baselines, **centre**: compares BOMVI to the baselines, **right**: compares BOMLA with different $\lambda$ values to BOMVI.

## 6.2 PENTATHLON: SEQUENTIAL DATASETS

We implement BOMLA and BOMVI to the pentathlon 5-way 1-shot classification sequence:

$$\text{Omniglot} \rightarrow \text{CIFAR-FS} \rightarrow \textit{mini}\text{ImageNet} \rightarrow \text{VGG-Flowers} \rightarrow \text{Aircraft}$$

The details of this experiment and the datasets can be found in Appendix C.2. We compare BOMLA and BOMVI to the baseline TOE, and running MAML continuously on the sequential datasets for meta-training.

Figure 2 shows that BOMLA and BOMVI are able to prevent few-shot catastrophic forgetting. TOE is also able to retain the few-shot performance as it has access to all datasets encountered so far. However, since it learns all datasets from random re-initialisation each time it encounters a new dataset, the meta-training time required to achieve a similarly good meta-evaluation performance is longer compared to other runs. The sequential MAML, on the other hand, catastrophically forgets the previously learned datasets but has the best performance on new datasets compared to other runs. TOE can be memory-intensive as the dataset sequence becomes longer. It takes the brute-force approach to prevent forgetting by memorising all datasets. Unlike TOE, our BOML approach only takes the posterior of the meta-parameters into consideration during optimisation. This gives a framework with an algorithmic complexity independent of the length of the dataset sequence.

@@@
New2: error-band

Tuning the posterior regulariser $\lambda$ mentioned in Section 4.2 corresponds to balancing between a smaller performance trade-off on a new dataset and less forgetting on previous datasets. As shown in **Appendix C.2 Figure 4**, a larger $\lambda = 1000$ results in a more concentrated Gaussian posterior and is therefore unable to learn new datasets well, but can better retain the performances on previous datasets. A smaller value $\lambda = 1$ on the other hand gives a widespread Gaussian posterior and learns better on new datasets by sacrificing the performance on the previous datasets. In this experiment, the value $\lambda = 100$ gives the best balance between old and new datasets. Ideally we seek for a good performance on both old and new datasets, but in reality there is a trade-off between retaining performance on old datasets and learning well on new datasets due to posterior approximation errors.

@@@
New2: $\lambda$-comparing plot in App. C.2

As shown in Figures 1 and 2, BOMLA with appropriate $\lambda$ is superior to BOMVI. This is due to BOMLA having a better posterior approximation than BOMVI. Whilst BOMLA has a Gaussian approximate posterior with block-diagonal precision, BOMVI uses a Gaussian mean-field approximate posterior. Trippe & Turner (2017) compared the performances of variational inference with different covariance structures, and discovered that variational inference with block-diagonal covariance performs worse than mean-field approximation. This is because the block-diagonal covariance in variational inference prohibits variance reduction methods such as local reparameterisation trick for Monte Carlo estimation. The variance of the Monte Carlo estimate has been proven problematic (Kingma et al., 2015; Trippe & Turner, 2017). We address this issue in Section 4.3 and Appendix B.2 specifically to the meta-learning setting by modifying the inner loop quick adaptation.

@@@
New

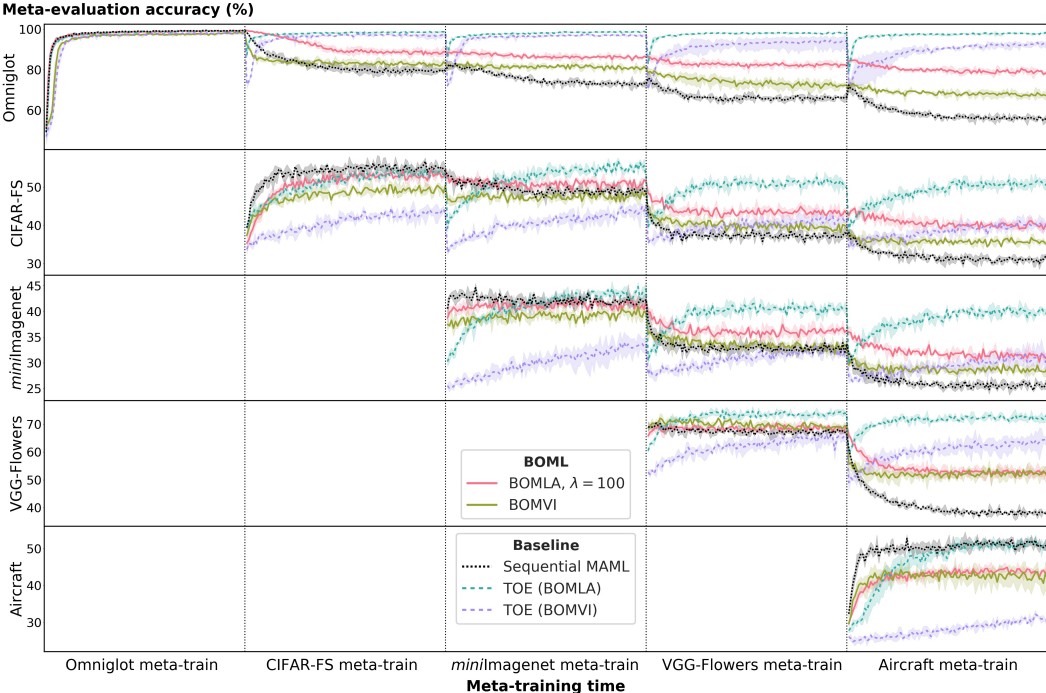

Figure 2: Meta-evaluation accuracy across 3 seed runs on each dataset along meta-training (**refer to Figure 3 for the enlarged version**). Higher accuracy values indicate better results with less forgetting as we proceed to new datasets. BOMLA with $\lambda = 100$ gives good performance in the off-diagonal plots (retains performances on previously learned datasets), and has a minor performance trade-off in the diagonal plots (learns less well on new datasets). Sequential MAML gives better performance in the diagonal plots (learns well on new datasets) but worse performance in the off-diagonal plots (forgets previously learned datasets). BOMVI is also able to retain performance on previous datasets, although it may be unable to perform as good as BOMLA due to sampling and estimator variance.

## 7 CONCLUSION

We introduced the Bayesian online meta-learning (BOML) framework with two algorithms: BOMLA and BOMVI. Our framework can overcome catastrophic forgetting in few-shot classification problems and can handle sequentially arriving few-shot tasks for online meta-learning. BOML merged the BOL framework and the MAML algorithm via Laplace approximation or variational continual learning. We proposed the necessary adjustments in the Hessian and Fisher approximation for BOMLA, as we are optimising the meta-parameters for few-shot classification instead of the usual model parameters in large-scale supervised classification. The experiments show that BOMLA and BOMVI are able to retain the few-shot classification ability when trained on sequential datasets with evident distributional shift, resulting in the ability to perform few-shot classification on multiple datasets with a single meta-learned model. BOMLA and BOMVI are also able to continually learn to few-shot classify novel tasks as the meta-training tasks arrive sequentially for learning.

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

# A BACKGROUND

This section provides a background explanation of using BOL to find the posterior of a model parameters and overcome catastrophic forgetting, commonly for large-scale supervised classification. We will then apply this approach to our recursion in Eq. (5).

The posterior is typically intractable due to the enormous size of the modern neural network architectures. This leads to the requirement for a good approximation of the posterior of the meta-parameters. A particularly suitable candidate for this purpose in meta-learning is the Laplace approximation (MacKay, 1992; Ritter et al., 2018b), as it simply adds a quadratic regulariser to the training objective. Variational inference is another possible method to obtain an approximation for the posterior of the meta-parameters.

## A.1 BAYESIAN ONLINE LEARNING

Upon the arrival of the new $\mathcal{D}_{t+1}$, we are interested in a MAP estimate $\theta^* = \arg\max_\theta p(\theta|\mathcal{D}_{1:t+1})$ for the parameters $\theta$ of a neural network. Using Bayes' rule on the posterior gives the recursive formula

$$p(\theta|\mathcal{D}_{1:t+1}) \propto p(\mathcal{D}_{t+1}|\theta)p(\theta|\mathcal{D}_{1:t}) \tag{16}$$

where Eq. (16) follows from the assumption that each dataset is independent given $\theta$. As the normalised posterior $p(\theta|\mathcal{D}_{1:t})$ is usually intractable, it may be approximated by a parametric distribution $q$ with parameter $\phi_t$. The BOL framework consists of the *update step* and the *projection step* (Opper, 1998). The update step uses the approximate posterior $q(\theta|\phi_t)$ obtained from the previous step for an update in the form of Eq. (16):

$$p(\theta|\mathcal{D}_{1:t+1}, \phi_t) \propto p(\mathcal{D}_{t+1}|\theta)q(\theta|\phi_t). \tag{17}$$

The new posterior $p(\theta|\mathcal{D}_{1:t+1}, \phi_t)$ might not belong to the same parametric family as $q(\theta|\phi_t)$. In this case, the new posterior has to be projected into the same parametric family to obtain $q(\theta|\phi_{t+1})$. Opper (1998) performs this projection by minimising the KL-divergence between the new posterior and the parametric $q$, while Ritter et al. (2018a) use the Laplace approximation and Nguyen et al. (2018) use variational inference.

## A.2 LAPLACE APPROXIMATION

We consider finding a MAP estimate following from Eq. (16):

$$\theta_{t+1}^* = \arg\max_\theta p(\theta|\mathcal{D}_{1:t+1}) = \arg\max_\theta \{\log p(\mathcal{D}_{t+1}|\theta) + \log p(\theta|\mathcal{D}_{1:t})\}. \tag{18}$$

Since the posterior $p(\theta|\mathcal{D}_{1:t})$ of a neural network is intractable except for small architectures, the unnormalised posterior $\tilde{p}(\theta|\mathcal{D}_{1:t})$ is considered instead. Performing Taylor expansion on the logarithm of the unnormalised posterior around a mode $\theta_t^*$ gives

$$\log \tilde{p}(\theta|\mathcal{D}_{1:t}) \simeq \log \tilde{p}(\theta|\mathcal{D}_{1:t})\big|_{\theta=\theta_t^*} - \frac{1}{2}(\theta - \theta_t^*)^T A_t (\theta - \theta_t^*), \tag{19}$$

where $A_t$ denotes the Hessian matrix of the negative log-posterior evaluated at $\theta_t^*$. The expansion in Eq. (19) suggests using a Gaussian approximate posterior. Given the parameter $\phi_t = \{\mu_t, \Lambda_t\}$, a mean $\mu_{t+1}$ for step $t+1$ can be obtained by finding a mode of the approximate posterior as follows via standard gradient-based optimisation:

$$\mu_{t+1} = \arg\max_\theta \log p(\mathcal{D}_{t+1}|\theta) - \frac{1}{2}(\theta - \mu_t)^T \Lambda_t (\theta - \mu_t). \tag{20}$$

The precision matrix is updated as $\Lambda_{t+1} = H_{t+1} + \Lambda_t$, where $H_{t+1}$ is the Hessian matrix of the negative log-likelihood for $\mathcal{D}_{t+1}$ evaluated at $\mu_{t+1}$ with entries

$$H_{t+1}^{ij} = -\frac{\partial^2}{\partial\theta^{(i)}\partial\theta^{(j)}} \log p(\mathcal{D}_{t+1}|\theta)\bigg|_{\theta=\mu_{t+1}}. \tag{21}$$

For a neural network model, gradient-based optimisation methods such as SGD (Robbins & Monro, 1951) and Adam (Kingma & Ba, 2015) are the standard gradient-based methods in finding a mode for the Laplace approximation in Eq. (20). We show in Section 4.1 that this provides a well-suited skeleton to implement Bayesian online meta-learning in Eq. (5) with the mode-seeking optimisation procedure.

## A.3 BLOCK-DIAGONAL HESSIAN APPROXIMATION

Since the full Hessian matrix in Eq. (21) is intractable for large neural networks, we seek for an efficient and relatively close approximation to the Hessian matrix. Diagonal approximations (Denker & LeCun, 1991; Kirkpatrick et al., 2017) are memory and computationally efficient, but sacrifice approximation accuracy as they ignore the interaction between parameters. Consider instead separating the Hessian matrix into blocks where different blocks are associated to different layers of a neural network. A particular diagonal block corresponds to the Hessian for a particular layer of the neural network. The block-diagonal Kronecker-factored approximation (Martens & Grosse, 2015; Grosse & Martens, 2016; Botev et al., 2017) utilises the fact that each diagonal block of the Hessian is Kronecker-factored for a single data point. This provides a better Hessian approximation as it takes the parameter interactions within a layer into consideration.

### A.3.1 KRONECKER-FACTORED APPROXIMATION

Consider a neural network with $L$ layers and parameter $\theta = [\text{vec}(W_1)^T, \ldots, \text{vec}(W_L)^T]^T$ where $W_\ell$ is the weight of layer $\ell$ for $\ell = \{1, \ldots, L\}$ and vec denotes stacking the columns of a matrix into a vector. We denote the input of the neural network as $a_0 = x$ and the output of the neural network as $h_L$. As the input passes through each layer of the neural network, we have the pre-activation for layer $\ell$ as $h_\ell = W_\ell a_{\ell-1}$ and the activation as $a_\ell = f_\ell(h_\ell)$ where $f_\ell$ is the activation function of layer $\ell$. If a bias vector is applicable in calculating the pre-activation of a layer, we append the bias vector to the last column of the weight matrix and append a scalar one to the last element of the activation. The gradient $g_\ell$ of loss $L_\theta(x, y) = -\log p(y|x, \theta)$ with respect to $h_\ell$ for an input-target pair $(x, y)$ is the pre-activation gradient for layer $\ell$.

Martens & Grosse (2015) show that the $\ell$-th diagonal block $F_\ell$ of the Fisher information matrix $F$ can be approximated by the Kronecker product between the expectation of the outer product of the $(\ell - 1)$-th layer activation and the $\ell$-th layer pre-activation gradient:

$$F_\ell = \mathbb{E}_{x,y}[a_{\ell-1}a_{\ell-1}^T \otimes g_\ell g_\ell^T] \tag{22}$$

$$\approx \mathbb{E}_x[a_{\ell-1}a_{\ell-1}^T] \otimes \mathbb{E}_{y|x}[g_\ell g_\ell^T] \tag{23}$$

$$= A_{\ell-1} \otimes G_\ell, \tag{24}$$

where $A_{\ell-1} = \mathbb{E}_x[a_{\ell-1}a_{\ell-1}^T]$ and $G_\ell = \mathbb{E}_{y|x}[g_\ell g_\ell^T]$. Grosse & Martens (2016) extend the block-diagonal Kronecker-factored Fisher approximation for fully-connected layers to that for convolution layers. The Gaussian log-probability term can be calculated efficiently without expanding the Kronecker product using the identity

$$(A_{\ell-1} \otimes G_\ell) \text{vec}(W_\ell - W_\ell^*) = \text{vec}(G_\ell(W_\ell - W_\ell^*)A_{\ell-1}^T). \tag{25}$$

As we mentioned in Section 4.2, approximating the Hessian with the one-step SGD inner loop assumption results in having terms that multiply two or more Kronecker products together. The $\ell$-th diagonal block of $\widetilde{F}$ in Eq. (11) is

$$\widetilde{F}_\ell = \frac{1}{M} \sum_{m=1}^{M} (I - A_{\ell-1}^m \otimes G_\ell^m)(\widetilde{A}_{\ell-1}^m \otimes \widetilde{G}_\ell^m)(I - A_{\ell-1}^m \otimes G_\ell^m)^T, \tag{26}$$

where $A_{\ell-1}^m \otimes G_\ell^m$ is the Kronecker product corresponding to the Hessian in Eq. (13) for task or batch $m$. We expand $\widetilde{F}_\ell$ using the Kronecker product property:

$$(A_{\ell-1}^m \otimes G_\ell^m)(\widetilde{A}_{\ell-1}^m \otimes \widetilde{G}_\ell^m) = A_{\ell-1}^m \widetilde{A}_{\ell-1}^m \otimes G_\ell^m \widetilde{G}_\ell^m. \tag{27}$$

This gives

$$\widetilde{F}_\ell = \frac{1}{M} \sum_{m=1}^{M} \left\{ \widetilde{A}_{\ell-1}^m \otimes \widetilde{G}_\ell^m - A_{\ell-1}^m \widetilde{A}_{\ell-1}^m \otimes G_\ell^m \widetilde{G}_\ell^m - \widetilde{A}_{\ell-1}^m (A_{\ell-1}^m)^T \otimes \widetilde{G}_\ell^m (G_\ell^m)^T \right.$$
$$\left. + A_{\ell-1}^m \widetilde{A}_{\ell-1}^m (A_{\ell-1}^m)^T \otimes G_\ell^m \widetilde{G}_\ell^m (G_\ell^m)^T \right\}. \tag{28}$$

Finally, moving the meta-batch (or batch) averaging into the Kronecker factors gives the approximation:

$$
\begin{aligned}
\widetilde{F}_\ell \approx & \widetilde{\boldsymbol{A}}_{\ell-1} \otimes \widetilde{\boldsymbol{G}}_\ell - \boldsymbol{A}_{\ell-1}\widetilde{\boldsymbol{A}}_{\ell-1} \otimes \boldsymbol{G}_\ell\widetilde{\boldsymbol{G}}_\ell - \widetilde{\boldsymbol{A}}_{\ell-1}(\boldsymbol{A}_{\ell-1})^T \otimes \widetilde{\boldsymbol{G}}_\ell(\boldsymbol{G}_\ell)^T \\
& + \boldsymbol{A}_{\ell-1}\widetilde{\boldsymbol{A}}_{\ell-1}(\boldsymbol{A}_{\ell-1})^T \otimes \boldsymbol{G}_\ell\widetilde{\boldsymbol{G}}_\ell(\boldsymbol{G}_\ell)^T,
\end{aligned}
\tag{29}
$$

where $\widetilde{\boldsymbol{A}}_{\ell-1} = \frac{1}{M}\sum_m \widetilde{A}^m_{\ell-1}$, $\widetilde{\boldsymbol{G}}_\ell = \frac{1}{M}\sum_m \widetilde{G}^m_\ell$, $\boldsymbol{A}_{\ell-1}\widetilde{\boldsymbol{A}}_{\ell-1} = \frac{1}{M}\sum_m A^m_{\ell-1}\widetilde{A}^m_{\ell-1}$, and so on.

### A.3.2 POSTERIOR REGULARISING HYPERPARAMETER FOR PRECISION UPDATE

Ritter et al. (2018a) use a hyperparameter $\lambda$ as a multiplier to the Hessian when updating the precision:

$$
\Lambda_{t+1} = \lambda H_{t+1} + \Lambda_t.
\tag{30}
$$

In the large-scale supervised classification setting, this hyperparameter has a regularising effect on the Gaussian posterior approximation for a balance between having a good performance on a new dataset and maintaining the performance on previous datasets (Ritter et al., 2018a). A large $\lambda$ results in a sharply peaked Gaussian posterior and is therefore unable to learn new datasets well, but can prevent forgetting previously learned datasets. A small $\lambda$ on the other hand gives a dispersed Gaussian posterior and allows better performance on new datasets by sacrificing the performance on the previous datasets.

### A.4 VARIATIONAL CONTINUAL LEARNING

The variational continual learning method (Nguyen et al., 2018) also provides a suitable meta-training framework for Bayesian online meta-learning in Eq. (5). Consider approximating the posterior $q$ by minimising the KL-divergence between the parametric $q$ and the new posterior as in the projection step in Eq. (17), where $q$ belongs to some pre-determined approximate posterior family $\mathcal{Q}$ with parameters $\phi_t$:

$$
q(\theta|\phi_{t+1}) = \underset{q \in \mathcal{Q}}{\arg\min}\, D_{\mathrm{KL}}(q(\theta|\phi)\|p(\mathcal{D}_{t+1}|\theta)q(\theta|\phi_t))
\tag{31}
$$

$$
= \underset{q \in \mathcal{Q}}{\arg\min}\, \big\{ -\mathbb{E}_{q(\theta|\phi)}[\log p(\mathcal{D}_{t+1}|\theta)] + D_{\mathrm{KL}}(q(\theta|\phi)\|q(\theta|\phi_t)) \big\}.
\tag{32}
$$

The optimisation in Eq. (32) leads to the objective

$$
\phi_{t+1} = \underset{\phi}{\arg\min}\, \big\{ -\mathbb{E}_{q(\theta|\phi)}[\log p(\mathcal{D}_{t+1}|\theta)] + D_{\mathrm{KL}}(q(\theta|\phi)\|q(\theta|\phi_t)) \big\}.
\tag{33}
$$

One can use a Gaussian mean-field approximate posterior $q(\theta|\phi_t) = \prod_{d=1}^{D} N(\mu_{t,d}, \sigma_{t,d}^2)$, where $\phi_t = \{\mu_{t,d}, \sigma_{t,d}\}_{d=1}^{D}$ and $D = \dim(\theta)$. The first term in Eq. (33) can be estimated via Monte Carlo with local reparameterisation trick (Kingma et al., 2015), and the second KL-divergence term has a closed form for Gaussian distributions.

## B  ALGORITHMS

### B.1  BOMLA AND BOMVI

Algorithm 1 gives the pseudo-code of the BOMLA algorithm for the sequential datasets setting, with the corresponding variation for the sequential tasks setting in brackets. The algorithm is formed of three main elements: meta-training on a specific dataset or task (line $4 - 11$), updating the Gaussian mean (line 12) and updating the Gaussian precision (line $13 - 16$). For the precision update, we approximate the Hessian using block-diagonal Kronecker-factored approximation (BD-KFA).

Algorithm 2 gives the pseudo-code of the BOMVI algorithm for the sequential datasets setting, with the corresponding variation for the sequential tasks setting in brackets. The algorithm is formed of two main elements: meta-training on a specific dataset or task (line $4 - 11$) and updating the parameters of the Gaussian mean-field approximate posterior (line 12).

---

**Algorithm 1** Bayesian online meta-learning with Laplace approximation (BOMLA)

---

1: **Require:** sequential datasets (or tasks) $\widetilde{\mathcal{D}}_1, \ldots, \widetilde{\mathcal{D}}_T$, learning rate $\alpha$, posterior regulariser $\lambda$, number of meta-training iterations (or epochs) $J$, meta-batch size (or number of batches) $M$
2: **Initialise:** $\mu_0, \Lambda_0, \theta$
3: **for** $t = 1$ **to** $T$ **do**
4:     **for** $i = 1, \ldots, J$ **do**         ▷ meta-training on dataset or task $\widetilde{\mathcal{D}}_t$ @@@ New: added $J$
5:         **for** $m = 1$ **to** $M$ **do**
6:             Sample task (or split the batch) $\widetilde{\mathcal{D}}_t^m = \widetilde{\mathcal{D}}_t^{m,S} \cup \widetilde{\mathcal{D}}_t^{m,Q}$
7:             Inner update $\tilde{\theta}^m = SGD_k(\mathcal{L}(\theta, \widetilde{\mathcal{D}}_t^{m,S}))$
8:         **end for**
9:         Evaluate loss $f_t^{\text{BOMLA}}(\theta, \mu_{t-1}, \Lambda_{t-1})$ in Eq. (8)
10:         Outer update $\theta \leftarrow \theta - \alpha \nabla_\theta f_t^{\text{BOMLA}}(\theta, \mu_{t-1}, \Lambda_{t-1})$
11:     **end for**
12:     Update mean $\mu_t \leftarrow \theta$         ▷ update posterior mean
13:     For sequential datasets, sample $M$ tasks for Hessian approximation
14:     Run inner update in line 7 for each task (or for each batch)
15:     Approximate $\widetilde{H}_t$ with BD-KFA to $\widetilde{F}$ in Eq. (11)
16:     Update precision $\Lambda_t \leftarrow \lambda \widetilde{H}_t + \Lambda_{t-1}$         ▷ update posterior precision
17: **end for**

---

**Algorithm 2** Bayesian online meta-learning with variational inference (BOMVI)

---

1: **Require:** sequential datasets (or tasks) $\widetilde{\mathcal{D}}_1, \ldots, \widetilde{\mathcal{D}}_T$, learning rate $\alpha$, number of meta-training iterations (or epochs) $J$, meta-batch size (or number of batches) $M$
2: **Initialise:** $\phi_0 = \{\mu_0, \sigma_0\}$
3: **for** $t = 1$ **to** $T$ **do**
4:     **for** $i = 1, 2, \ldots, J$ **do**         ▷ meta-training on dataset or task $\widetilde{\mathcal{D}}_t$ @@@ New: added $J$
5:         **for** $m = 1$ **to** $M$ **do**
6:             Sample task (or split the batch) $\widetilde{\mathcal{D}}_t^m = \widetilde{\mathcal{D}}_t^{m,S} \cup \widetilde{\mathcal{D}}_t^{m,Q}$
7:             Inner update $\tilde{\theta}^m = SGD_k(\mathcal{L}(\theta, \widetilde{\mathcal{D}}_t^{m,S}))$
8:         **end for**
9:         Evaluate loss $f_t^{\text{BOMVI}}(\phi, \phi_{t-1})$ in Eq. (15)
10:         Outer update $\mu \leftarrow \mu - \alpha \nabla_\mu f_t^{\text{BOMVI}}(\phi, \phi_{t-1})$, and $\sigma \leftarrow \sigma - \alpha \nabla_\sigma f_t^{\text{BOMVI}}(\phi, \phi_{t-1})$
11:     **end for**
12:     Update $\mu_t \leftarrow \mu$ and $\sigma_t \leftarrow \sigma$         ▷ update posterior parameters
13: **end for**

---

### B.2 BOMVI MONTE CARLO ESTIMATOR

Recall that the BOMVI objective is:

$$f_{t+1}^{\text{BOMVI}}(\phi, \phi_t) = -\frac{1}{M} \sum_{m=1}^{M} \mathbb{E}_{q(\theta|\phi)} \big[ \log p(\widetilde{\mathcal{D}}_{t+1}^{m,Q}|\tilde{\theta}^m) \big] - \frac{1}{M} \sum_{m=1}^{M} \mathbb{E}_{q(\theta|\phi)} \big[ \log p(\widetilde{\mathcal{D}}_{t+1}^{m,S}|\theta) \big]$$
$$+ D_{\text{KL}}(q(\theta|\phi)\|q(\theta|\phi_t)),$$

where $\tilde{\theta}^m = SGD_k(\mathcal{L}(\theta, \widetilde{\mathcal{D}}_{t+1}^{m,S}))$ for $m = 1, \ldots, M$. The Monte Carlo estimator for the first term of the BOMVI objective is difficult to compute, as every sampled meta-parameters $\theta_r$ for $r = 1, \ldots, R$ has to undergo a few-shot quick adaptation prior to the log-likelihood evaluation. As a consequence the estimator is prone to a large variance. Moreover, every quickly-adapted sample $\theta_r$ contributes to the meta-learning gradients of the posterior mean and covariance, resulting in a high computational cost when taking the meta-gradients.

To solve these impediments, we introduce a slight modification to the SGD quick adaptation $\tilde{\theta}^m$. Instead of taking the gradients with respect to the sampled meta-parameters, we consider the gradients with respect to the *posterior mean*. A one-step SGD quick adaptation, for instance, becomes:

$$\tilde{\theta}^m = \theta - \alpha \nabla_{\mu_t} \mathcal{L}(\mu_t, \widetilde{\mathcal{D}}_{t+1}^{m,S}). \tag{34}$$

This gives $\widetilde{\theta}^m \sim N(\widetilde{\mu}_t, \mathrm{diag}(\sigma_t^2))$ where

$$\widetilde{\mu}_t = \mu_t - \alpha \nabla_{\mu_t} \mathcal{L}(\mu_t, \widetilde{\mathcal{D}}_{t+1}^{m,S}), \tag{35}$$

since $\theta \sim N(\mu_t, \mathrm{diag}(\sigma_t^2))$. A quick adaptation with more steps works in a similar fashion. With this modification, we can calculate the Monte Carlo estimator for the first term using the local reparameterisation trick as usual.

## C   EXPERIMENTS

### C.1   OMNIGLOT: SEQUENTIAL TASKS

In this experiment, we use the model architecture proposed by Vinyals et al. (2016) that takes 4 modules with 64 filters of size $3 \times 3$, followed by a batch normalisation, a ReLU activation and a $2 \times 2$ max-pooling. A fully-connected layer is appended to the final module before getting the class probabilities with softmax. Table 1 shows the hyperparameters used in this experiment.

The Omniglot dataset comprises 50 alphabets (super-classes). Each alphabet has numerous characters (classes) and each character has 20 instances. As the meta-training alphabets arrive sequentially, we form **non-overlapping** sequential tasks from each arriving alphabet, and the tasks also do not overlap in the characters. We use 35 alphabets for meta-training, 7 alphabets for validation and 8 alphabets for meta-evaluation. The alphabet splits are as follows:

35 alphabets for meta-training:

```
Kannada, Burmese_(Myanmar), Malay_(Jawi_-_Arabic), Grantha,
Atlantean, Ojibwe_(Canadian_Aboriginal_Syllabics), Balinese,
Japanese_(katakana), Hebrew, Japanese_(hiragana), Keble,
'Old_Church_Slavonic_(Cyrillic), Asomtavruli_(Georgian),
Tengwar, Aurek-Besh, Sanskrit, Manipuri, Early_Aramaic, Oriya,
Mongolian, Avesta, Malayalam, Tifinagh, Angelic, Latin, Braille,
Inuktitut_(Canadian_Aboriginal_Syllabics), Alphabet_of_the_Magi,
Armenian, Korean, Gurmukhi, ULOG, Bengali, Gujarati, Sylheti
```

7 alphabets for validation:

```
Ge_ez, Cyrillic, Glagolitic, N_Ko, Arcadian, Anglo-Saxon_Futhorc,
Blackfoot_(Canadian_Aboriginal_Syllabics)
```

8 alphabets for meta-evaluation:

```
Syriac_(Serto), Atemayar_Qelisayer, Tibetan, Futurama,
Mkhedruli_(Georgian), Syriac_(Estrangelo), Tagalog, Greek
```

Table 1: Hyperparameters for the Omniglot sequential tasks experiment

| Hyperparameter | BOMLA | BOMVI |
|---|---|---|
| Posterior regulariser $\lambda$ | 0.1 | - |
| Precision initialisation values | $10^{-4} \sim 10^{-2}$ | - |
| Covariance initialisation values | - | $\exp(-10)$ |
| Number of Monte Carlo samples | - | 5 |
| Number of batch $M$ | 1 | 1 |
| Number of query samples per class (meta-evaluation) | 15 | 15 |
| Number of epochs per task | 50 | 50 |
| Number of inner SGD steps in meta-training ($k$) | 5 | 5 |
| Inner SGD learning rate ($\alpha$) | 0.1 | 0.1 |
| Outer loop optimiser | Adam | Adam |
| Outer loop learning rate | 0.001 | 0.001 |
| Number of tasks sampled for meta-evaluation | 100 | 100 |
| Number of inner SGD steps in meta-evaluation ($k$) | 10 | 10 |

## C.2 PENTATHLON: SEQUENTIAL DATASETS

We use the model architecture proposed by Vinyals et al. (2016) in this experiment, as we did for the sequential tasks experiment. Tables 2 and 3 are the hyperparameters used in this experiment.

**Omniglot:**    The Omniglot dataset (Lake et al., 2011) comprises 1623 characters from 50 alphabets and each character has 20 instances. New classes with rotations in the multiples of $90°$ are formed after splitting the classes for meta-training, validation and meta-evaluation. We use 1100 characters for meta-training, 100 characters for validation and the remaining for meta-evaluation.

**CIFAR-FS:**    The CIFAR-FS dataset (Bertinetto et al., 2019) has 100 classes of objects and each class comprises 600 images. We use the same split as Bertinetto et al. (2019): 64 classes for meta-training, 16 classes for validation and 20 classes for meta-evaluation.

***mini*ImageNet:**    The *mini*ImageNet dataset (Vinyals et al., 2016) takes 100 classes and 600 instances in each class from the ImageNet dataset. We use the same split as Ravi & Larochelle (2017): 64 classes for meta-training, 16 classes for validation and 20 classes for meta-evaluation.

**VGG-Flowers:**    The VGG-Flowers dataset (Nilsback & Zisserman, 2008) comprises 102 different types of flowers as the classes. This dataset has 8,189 instances in total. We randomly split 66 classes for meta-training, 16 classes for validation and 20 classes for meta-evaluation.

**Aircraft:**    The Aircraft dataset (Maji et al., 2013) is a fine-grained dataset consisting of 100 different aircraft models as the classes and each class has 100 instances. We randomly split 64 classes for meta-training, 16 classes for validation and 20 classes for meta-evaluation.

Table 2: Hyperparameters for the pentathlon experiment (same value for all datasets)

| Hyperparameter | BOMLA | BOMVI |
|---|---|---|
| Posterior regulariser $\lambda$ | (various values) | - |
| Precision initialisation values | $10^{-4} \sim 10^{-2}$ | - |
| Number of tasks sampled for Hessian approx. | 5000 | - |
| Covariance initialisation values | - | $\exp(-5)$ |
| Number of Monte Carlo samples | - | 20 |
| Meta-batch size $M$ | 32 | 32 |
| Number of query samples per class | 15 | 15 |
| Number of iterations per dataset | 5000 | 5000 |
| Outer loop optimiser | Adam | Adam |
| Outer loop learning rate | 0.001 | 0.001 |
| Number of tasks sampled for meta-evaluation | 100 | 100 |

Table 3: Hyperparameters for the pentathlon sequential datasets experiment (individual datasets)

| Hyperparameter | Omniglot | CIFAR-FS | *mini*ImageNet | VGG-Flowers | Aircraft |
|---|---|---|---|---|---|
| Number of inner SGD steps in meta-training ($k$) | 1 | 5 | 5 | 5 | 5 |
| Inner SGD learning rate ($\alpha$) | 0.4 | 0.1 | 0.1 | 0.1 | 0.1 |
| Outer learning rate decay schedule | - | $\times 0.1$ halfway | $\times 0.1$ halfway | $\times 0.1$ per 1000 iterations | $\times 0.1$ halfway |
| Number of inner SGD steps in meta-evaluation | 3 | 10 | 10 | 10 | 10 |

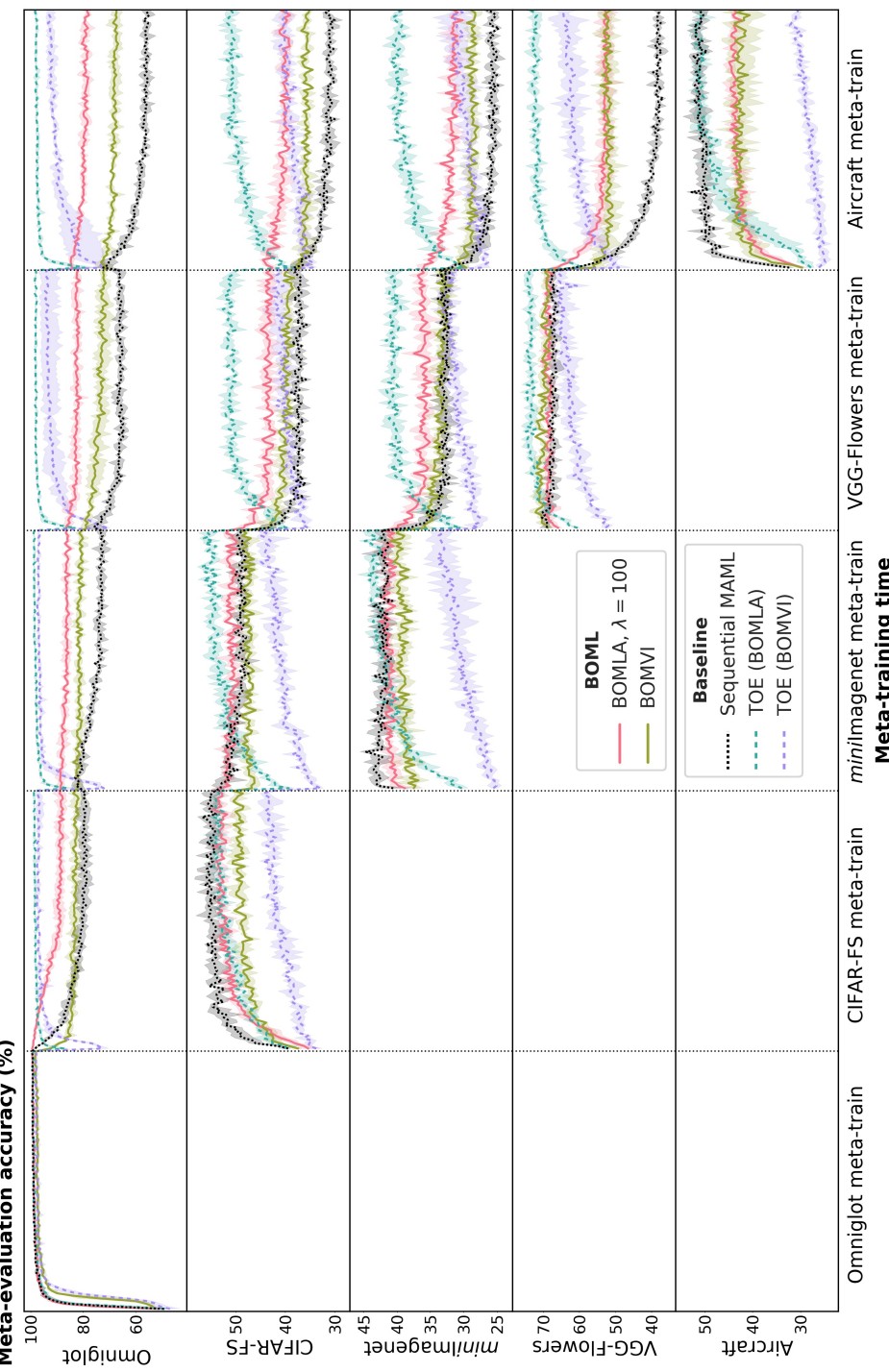

Figure 3: Meta-evaluation accuracy across 3 seed runs on each dataset along meta-training. Higher accuracy values indicate better results with less forgetting as we proceed to new datasets. BOMLA with $\lambda = 100$ gives better performance in the off-diagonal plots (retains performances on previously learned datasets), and has a minor performance trade-off in the diagonal plots (learns less well on new datasets). Sequential MAML gives better performance in the diagonal plots (learns well on new datasets) but worse performance in the off-diagonal plots (forgets previously learned datasets). BOMVI is also able to retain performance on previous datasets, although it may be unable to perform as good as BOMLA due to sampling and estimator variance.

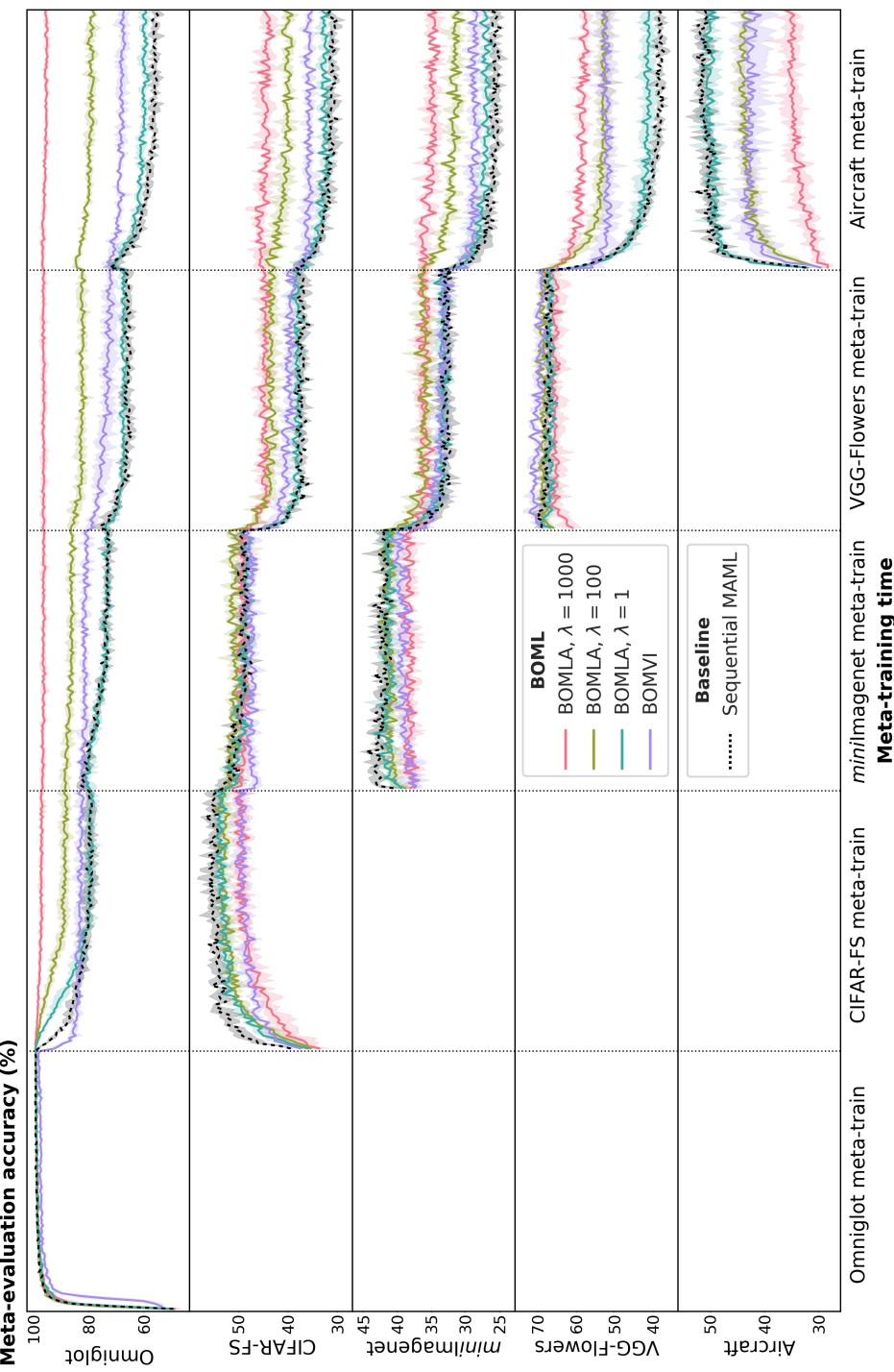

Figure 4: Meta-evaluation accuracy across 3 seed runs on each dataset along meta-training. Higher accuracy values indicate better results with less forgetting as we proceed to new datasets. BOMLA with a large $\lambda = 1000$ gives better performance in the off-diagonal plots (retains performances on previously learned datasets) but worse performance in the diagonal plots (does not learn well on new datasets). A small $\lambda = 1$ gives better performance in the diagonal plots (learns well on new datasets) but worse performance in the off-diagonal plots (forgets previously learned datasets). BOMVI is also able to retain performance on previous datasets, although it may be unable to learn new datasets as good as BOMLA due to sampling and estimator variance.

