# OpenReview forum: "Bayesian Online Meta-Learning"
_ICLR.cc/2021/Conference — Reject_

### Official Review · AnonReviewer4 · 2020-10-28
**Combination of Bayesian online learning and meta-learning - neat formulation but missing key references and baselines**

**Rating:** 7
**Confidence:** 4

**Review:**

The paper proposes an Bayesian approach to online meta-learning. This is done by lifting (approximate) sequential Bayesian inference from the model parameters to the meta-parameters. Two approaches are proposed to do this: (i) Laplace Approximation (LA), thereby extending Ritter et al’s method from online learning to online meta-learning; and (ii) VI, thereby extending Nguyen et al’s Variational Continual Learning (VCL) method in a similar way. Experiments are performed by converting existing meta-learning benchmarks into online settings in two ways, which they call “sequential tasks” and “sequential datasets” respectively. The experimental results show resistance to catastrophic forgetting in both of these experimental settings.

In general, the novelty is somewhat limited, as this extension of LA/VCL to meta learning is similar in principle to the Grant et al formulation as a hierarchical Bayesian model. The main difference  is that the inner loop is done using the standard fast adaptation method from MAML rather than using LA/VI.

If you’re only using a point estimate (“We are interested in a MAP estimate $\theta^*$”) then this is not truly Bayesian online learning. I assume that the choice of doing it this way is to sidestep the difficulty of integrating over the parameters to form the posterior predictive, but loses the advantages of having a Bayesian posterior.

Missing references:
- ML-PIP/VERSA (Gordon et al 2019, Meta Learning Probabilistic Inference for Prediction, ICLR). This does Bayesian meta learning by directly optimising the posterior predictive, rather than the usual posterior in VI. It also makes use of amortisation networks so that at test time only forward passes through NNs are required. They don’t provide an online formulation. However, The posterior predictives can be used as prior predictives for future rounds of inference.
- BMAML (Kim et al 2018, Bayesian model-agnostic meta-learning, NeurIPS). This uses Stein Variational Gradient Descent (SVGD) to obtain the task posterior for a novel task, casting MAML as a hierarchical probabilistic model.
- Jerfel et al 2019, Reconciling meta-learning and continual learning with online mixtures of tasks, NeurIPS. This paper also uses a hierarchical Bayesian formulation, but takes a non parametric mixture approach using Dirichlet processes. They handle distribution shift in the online meta-learning setting by modelling latent task structure.

Baselines:
- The baselines in this paper are a bit weak. The two methods ML-PIP and BMAML mentioned above, along with the Grant et al hierarchical Bayes method, could all be applied here, but since they weren’t originally formulated for the setting it would require extra work to do so. However, several methods were mentioned in the related work setting which should have been included: the He et al Task agnostic continual learning and Harrison et al continuous meta learning methods.

Edit post reviewer responses:
- I had some misunderstandings in the review above, which have mostly been cleared up. I’ve raised my score accordingly

---

> ### Author Response · Authors · 2020-11-14
> **Response to Reviewer 4 (Part 1)**
>
> Thank you for your valuable feedback. We address your comments below in detail:
>
> > “...the novelty is somewhat limited, as this extension of LA/VCL to meta learning is similar in principle to the Grant et al formulation as a hierarchical Bayesian model.”
>
>
> Although our framework is related to Bayesian online learning (BOL), Laplace approximation (LA) and variational continual learning (VCL) in terms of concept, we emphasize that our BOML framework is a non-trivial extension from these works. Further derivation on the Hessian approximation and re-defining the posterior objective are essential to arrive at the BOML framework. Below are our contributions in detail:
>
> + We derive a Hessian approximation in Section 3.2 and Appendix A.3.1 for the gradient-based meta-learning setting, which does not follow trivially from Martens and Grosse (2015). We further introduced the approximation required to consider the Hessian w.r.t. task-specific parameters $\tilde{\theta}$, and detailed the calculations in Appendix A.3.1.
>
> + We also derive the framework in Eq.(5) starting from the BOL principle in Eq.(16) using the support-query split and hierarchical task-parameters $\tilde{\theta}$, so that our framework is grounded based on Bayesian principles. Otherwise having only the hierarchical structure alone between meta-parameters $\theta$ and task-parameters $\tilde{\theta}$ would not be sufficient to give a strong-performing few-shot learning model that holds the knowledge learned in its posterior.
>
> + We would also like to emphasize that our work is very different from Grant et al. (2018) in terms of the end goal and context. Our work enables sequential meta-learning using Bayesian online learning, with the end goal of being able to perform few-shot learning on unseen tasks that are from multiple dataset domains (in the sequential datasets setting). Grant et al. (2018) also requires all base classes to be readily available in a batch for tasks sampling at each iteration, whereas ours allows meta-training on sequential tasks.
>
>
> > “... If you’re only using a point estimate (“We are interested in a MAP estimate ”) then this is not truly Bayesian online learning.”
>
> + We appreciate that you point this out and give us a chance to clarify. Our work clearly belongs to the Bayesian online learning conglomerate. Opper (1998) developed Bayesian online learning as a framework that considers the posterior of model parameters given the sequence of data $\mathcal{D}_{1:t+1}$, which is the very first principle that we use for the framework derivation in Eq.(5).
>
> + To clarify on MAP estimate, we first spell out MAP estimate $\theta^{*} = \arg \max_{\theta} p(\theta | \widetilde{\mathcal{D}} _ {1:t+1})$. A MAP estimate considers the posterior of the meta-parameters $\theta$, given all datasets or tasks $\widetilde{\mathcal{D}} _ 1, \ldots, \widetilde{\mathcal{D}} _ {t+1}$ encountered (sequentially) so far. In other words, this objective seeks for a meta-parameters posterior that 'performs well' on all sequential datasets or tasks. In this paper, we define a 'well-performing' meta-parameters posterior as being able to do few-shot classification on unseen tasks. In sequential datasets setting, our meta-parameters posterior aims to be able to perform few-shot classification on unseen tasks from **all dataset domains** encountered (Omniglot, CIFAR-FS, *mini*Imagenet, VGG-Flowers, Aircraft). In sequential tasks setting, the meta-parameters posterior aims to few-shot classify on unseen tasks from **one dataset domain** (Omniglot in our experiment).
>
> > “... ML-PIP/VERSA don’t provide an online formulation. However, The posterior predictives can be used as prior predictives for future rounds of inference. ... I assume that the choice of doing it this way is to sidestep the difficulty of integrating over the parameters to form the posterior predictive, but loses the advantages of having a Bayesian posterior.”
>
>
> While using the posterior predictive as the prior predictive for future rounds may be a sound idea, there is no formal recursive formulation that justifies such setting and usage. Our approach using a MAP estimate with posterior approximation originates from a grounded framework derivation in the recursion Eq.(5). Finding a posterior predictive may be a possible alternative, but we believe this does not degrade the MAP estimate method.

---

> > ### Author Response · Authors · 2020-11-14
> > **Response to Reviewer 4 (Part 2)**
> >
> > > Missing references
> >
> > The references to ML-PIP/Versa, BMAML and  Jerfel et al. (2019) will be added to the related work section in the paper, and the revised paper will very soon be updated.
> >
> > > Baselines
> >
> > While extending ML-PIP, BMAML and Grant et al. (2018) to an online setting seems to be an interesting direction for future research, we emphasize that this would not be a trivial extension. ML-PIP, BMAML and Grant et al. (2018) all requires sampling a meta-batch of few-shot tasks at each meta-training iteration, therefore unable to perform sequential meta-learning where the few-shot tasks arrive in an online manner.
> >
> >
> > We hope this justifies the novelty of our paper and clears your doubts. We once again thank you for your comments, and hope that you would consider raising your rating score.
> >
> >
> > ---
> >
> > **References**
> >
> > E. Grant, C. Finn, S. Levine, T. Darrell, and T. Griffiths. Recasting Gradient-Based Meta-Learning as Hierarchical Bayes. In International Conference on Learning Representations, 2018.
> >
> > G. Jerfel, E. Grant, T. Griffiths, and K. A. Heller. Reconciling Meta-Learning and Continual Learning with Online Mixtures of Tasks. In Advances in Neural Information Processing Systems 32, 2019.
> >
> > J. Martens and R. Grosse. Optimizing Neural Networks with Kronecker-Factored Approximate Curvature. In Proceedings of the 32nd International Conference on Machine Learning, 2015.
> >
> > M. Opper. A Bayesian Approach to Online Learning. Cambridge University Press, 1998.

---

> > > ### Comment · AnonReviewer4 · 2020-11-20
> > > **Response to Author Responses**
> > >
> > > Thank-you for the extensive response. I will retract my comments about the baselines, as I feel you have adequately answered my concerns there. I can also see more clearly the contributions, but if accepted it will help for the camera ready to make this clearer.
> > >
> > > I think I understand the point about the MAP estimate now, although going back and reading the part of the paper I can see where I was led astray. Basically, you’re looking for the overall MAP over all tasks, but in order to achieve that, you’ll keep updating an approximate posterior in an online fashion, in the hope that it gives you a good overall MAP estimate. I suppose something that’s hidden in this is that the actual posterior over $\theta$ isn’t terrible useful once you’re done, since you only want the MAP. However, perhaps there are uses, such as quantifying the uncertainty in the meta-learning process, or combining meta-posteriors for some sort of meta-meta-learning (!).
> > >
> > > I will update my review and score accordingly

---

### Official Review · AnonReviewer3 · 2020-10-29
**Review of Bayesian Online Meta-Learning**

**Rating:** 5
**Confidence:** 4

**Review:**

This work proposes a Bayesian approach to meta-learning from sequential data. Two algorithms are proposed. The first is based on the Laplace approximation to the model posterior which is made tractable by using K-FAC approximation of the Hessian. The second approaches uses a variational approximation for the posterior, where meta-learning corresponds to learning the variational prior. The experiments present results of the proposed method in sequential Omniglot and a pentathlon task involving different datasets.

The major drawback of the paper is a lack of description of the precise problem setting. Furthermore, certain motivations and comments appear to be unsubstantiated, and it would be great for the authors to provide rationale for them. Overall, in the current form, I find it hard to recommend the paper for acceptance. However, I am willing to reconsider based on author response to the questions.

**What is the setting of the paper?**
The paper over-emphasizes the sequential nature of the setting, but does not provide clear problem setup beyond this. There are multiple well-defined problem setups that all fall under the purview of sequential data. Some examples are:

1. *Regret minimization* : Here, no (distributional) assumptions about the tasks/datasets are made whatsoever. The goal is to compete with the best (meta) learner in hindsight. Cesa-Bianchi and Lugosi textbook would be a classic reference and Finn et al. 2019 (FTML) is a modern take in the context of meta-learning.

2. *i.i.d. datasets* : Here, tasks come from the same underlying distribution, just sequentially one after the other. In contrast to above, this places a stronger assumption about the world, but as a result may be able to develop more specialized algorithms. Note that regret minimization algorithms can be used for i.i.d. datasets (typically called online to batch conversion), while the reverse is generally not true.

It is unclear which setting the paper studies. My understanding is that setting 2 is more relevant to the paper, but the authors should clarify and explain the setting of the paper explicitly. I also suggest the authors make Section 2 to be related works and explain settings that are already known, in order to better position the contribution relative to literature. This also ensures better dissemination of scientific information.

**Why should we learn the MAP?**

The paper in Section 2 and 3 sets up an objective that aims to find the MAP estimate, i.e. $\theta^* = \arg \max_\theta P(\theta | \tilde{D}_{1:t+1})$. However it is not clear why this should be the objective. In a streaming setting, doing well on the past need not imply doing well on future streams of data unless some assumptions are made about the data distribution.

**Why cant we store the data in a "replay buffer"?**

The paper in multiple places states that storing historical data increases the algorithmic complexity (e.g. for FTML). Is this really a bottleneck? Can you present data or experiments to back this up -- my guess is that for the tasks/experiments considered in this work, storing historical data in memory is not that expensive. Furthermore, disk/memory costs are cheap and it is routine in ML these days to work with very large datasets. For example, deep RL stores all historical data in the form of a replay buffer for learning. Thus, it is unclear to me why FTML has not been considered for comparison?

---

> ### Author Response · Authors · 2020-11-14
> **Response to Reviewer 3 (Part 1)**
>
> Thank you for your engagement in suggesting improvements for our paper! We address your comments below in detail:
>
> > “... lack of description of the precise problem setting. ... What is the setting of the paper?”
>
>
> Many thanks for pointing out the setting clarity issue. You are right that setting 2 (i.i.d datasets) is more applicable in this paper. Below we explain our settings in detail:
>
> **Sequential datasets setting**
>
> As explained in Section 2.2, when a dataset $\mathcal{D}_i$ arrives it will be split into the meta-training dataset $\widetilde{\mathcal{D}}_i$ and the meta-evaluation dataset $\widehat{\mathcal{D}}_i$. The few-shot tasks sampled for meta-training and meta-evaluation are assumed to be drawn from the distribution $p(\mathcal{T}_i)$. This means each dataset $\mathcal{D}_1, \ldots, \mathcal{D}_t$ in the sequence is associated to the underlying few-shot task distributions $p(\mathcal{T}_1), \ldots, p(\mathcal{T}_t)$ respectively.
>
> **Sequential few-shot tasks setting**
>
> In this setting, a dataset $\mathcal{D}$ (say Omniglot) is being split into the meta-training set $\widetilde{\mathcal{D}}$ and the meta-evaluation set $\widehat{\mathcal{D}}$. The non-overlapping few-shot tasks $\widetilde{\mathcal{D}}_1, \ldots, \widetilde{\mathcal{D}}_t$ are generated from $\widetilde{\mathcal{D}}$. These few-shot tasks along with the tasks sampled from $\widehat{\mathcal{D}}$ during meta-evaluation are all drawn from the same task distribution $p(\mathcal{T})$.
>
>
> We will include this explanation to clarify our settings in Section 2.2, and will very soon update the revised paper.
>
>
> > “I also suggest the authors make Section 2 to be related works and explain settings that are already known, in order to better position the contribution relative to literature.”
>
> Thank you for your constructive suggestion. We have previously considered putting the related works to Section 2. However we find that starting with the BOML overview gives a better flow for the paper, since our BOML framework is not built up upon recent meta-learning methods other than MAML. Nonetheless, we appreciate this valuable suggestion. We will add an explanation on our problem settings to Section 2.2, and also explain the existing settings in the related work section. We will very soon revise our paper to include these details.
>
>
> > “Why should we learn the MAP?
> > The paper in Section 2 and 3 sets up an objective that aims to find the MAP estimate. However it is not clear why this should be the objective. In a streaming setting, doing well on the past need not imply doing well on future streams of data unless some assumptions are made about the data distribution.”
>
> + Overall, we agree that the data distribution assumption has to be made clearer in the paper (as explained in the first comment earlier). Kindly anticipate a revised version of the paper very soon with the assumptions on the data distributions made clear.
>
> + In order to clarify the usage of a MAP estimate, we first spell out MAP estimate $\theta^{*} = \arg \max_{\theta} p(\theta | \widetilde{\mathcal{D}} _ {1:t+1})$. A MAP estimate considers the posterior of the meta-parameters $\theta$, given all datasets or tasks $\widetilde{\mathcal{D}} _ 1, \ldots, \widetilde{\mathcal{D}} _ {t+1}$ encountered (sequentially) so far. In other words, this objective seeks for a meta-parameters posterior that 'performs well' on all sequential datasets or tasks. In this paper, we define a 'well-performing' meta-parameters posterior as being able to do few-shot classification on unseen tasks. In sequential datasets setting, our meta-parameters posterior aims to be able to perform few-shot classification on unseen tasks from **all dataset domains** (Omniglot, CIFAR-FS, *mini*Imagenet, VGG-Flowers and Aircraft) encountered so far. In sequential tasks setting, the meta-parameters posterior aims to few-shot classify on unseen tasks from only **one dataset domain** (Omniglot in our experiment).

---

> > ### Author Response · Authors · 2020-11-14
> > **Response to Reviewer 3 (Part 2)**
> >
> > > “Why cant we store the data in a "replay buffer"? The paper in multiple places states that storing historical data increases the algorithmic complexity (e.g. for FTML). Is this really a bottleneck? ... my guess is that for the tasks/experiments considered in this work, storing historical data in memory is not that expensive. Furthermore, disk/memory costs are cheap and it is routine in ML these days to work with very large datasets. ... Thus, it is unclear to me why FTML has not been considered for comparison?”
> >
> > **Replay buffer**
> >
> > In sequential datasets setting, storing a replay buffer would require storing the datasets as they arrive for meta-training. It is still possible to do so in our experiment setting, as there are only 5 datasets in the sequence. In fact, we have implemented this on the Train-on-everything (TOE) baseline in our experiments. It stores data buffer by memorising the datasets encountered so far. However if we proceed to a longer sequence of large datasets, say 100 ImageNet-alike datasets, it would be highly undesirable to keep all the data. In essence we aim to have meaningful updates on the model parameters using the data that arrive sequentially, so that the datasets or tasks learned previously can later be useful for few-shot learning on unseen tasks from the same domain. Keeping the data is surely a method for future retrieval, but this would defeat the purpose of doing an online learning, which by definition means to update the model each round using only the new data encountered.
> >
> > **Bottleneck and comparison**
> >
> > We would like to draw your attention to the differences between our work and FTML (Finn et al., 2019). Both works may be slightly similar in the online setting, but our work is rather different in end goal and implementation method. Below we address the bottleneck issue and comparison to FTML in detail:
> >
> > + The goal of FTML is to be able to decrease the number of training examples needed each round, as the tasks arrive sequentially for training. Our method focuses on being able to perform few-shot learning on unseen tasks (5-shot in sequential task Omniglot experiment and 1-shot in pentathlon), whereas FTML is not designed for few-shot learning in that sense.
> >
> > + FTML has to keep a task buffer that accumulates data from all tasks seen so far:
> >
> >     **sequential few-shot tasks setting**
> >
> >     This may be reasonable for the sequential few-shot tasks setting if the sequence is short. For instance, the longest task sequence that Finn et al. (2019) has in the experiments is 90 tasks, whereas our Omniglot experiment has more than 250 tasks.
> >
> >     **sequential datasets setting**
> >
> >     Keeping a replay buffer would be undesirable in the sequential datasets setting, because in practice it is often not possible to store all of the data. In our pentathlon experiment, having a replay buffer would means having to keep Omniglot, CIFAR-FS, *mini*Imagenet, VGG-Flowers, Aircraft as they arrive for training. In fact, the baseline comparison 'Train-on-everything' (TOE) in our experiments does indeed keep all the datasets. Keeping for 5 datasets as in our experiment is still manageable since these are relatively small academic benchmark datasets, but it would not be manageable if we have a long sequence of datasets that are as large as ImageNet. Ultimately we do not want to restrict our model in terms of the number or size of datasets it can learn from, as this would defeat the purpose of doing an online learning.
> >
> > + Explicitly keeping a memory on previous data often triggers an important question: how should the carried-forward data be processed in future task rounds, in order to accumulate knowledge? In Finn et al. (2019), few-shot tasks are sampled at each iteration from the task buffer that contains previous data. This contradicts the purpose of doing an online learning, which by definition means to update the model each round using only the new data encountered. If we have to re-train on previous data to avoid forgetting them, the training time may increase with the amount of data. We can of course clamp the amount of data at some maximal limit and perform sampling, but the final performance of such algorithm would be dependent on the samples being informative and of good quality which may vary across different seed runs. In contrast to memorising the datasets, having an implicit memory via the posterior automatically deals with the question on how to process carried-forward data and allows a better carry forward in previous experiences.
> >
> >
> > Nonetheless, FTML will be added to our experiments as a baseline comparison, but we emphasize that our method will still be the best performing method under the few-shot learning setting, because our framework is designed for few-shot purpose (5-shot for Omniglot and 1-shot for pentathlon) whilst FTML needs about 100 datapoints or more to achieve a good performance in their rainbow MNIST experiment.

---

> > > ### Author Response · Authors · 2020-11-14
> > > **Response to Reviewer 3 (Part 3)**
> > >
> > > We hope this clarifies the problem settings in our paper and clears your doubts. We once again thank you for your constructive comments, and hope that you would consider raising your score.
> > >
> > > ---
> > >
> > > **References**
> > > C. Finn, A. Rajeswaran, S. Kakade, and S. Levine. Online Meta-Learning. In Proceedings of the 36th International Conference on Machine Learning, 2019.

---

### Official Review · AnonReviewer2 · 2020-10-29
**The paper develops a semi-online Bayesian approach to meta-learning. The framework is interesting and tackles an important problem. My key concern is incermentality WRT previous work.**

**Rating:** 6
**Confidence:** 4

**Review:**

The paper develops a semi-online Bayesian approach to meta-learning, where tasks arrive sequentially and learning within any task is performed in batch mode (hence my terminology semi-online). It suggests a sequential between-task Bayesian update, eq. 5, and proposed three approximations to aid computation. The basic setup is motivated within the recently introduced MAML framework where the learning takes place by adapting a within-task parameter to effectively set up learning within each individual task, allowing the learner to transfer information between tasks, while remaining adaptive to a specific novel task. The authors phrase this idea in the Bayesian language of posterior distributions, that are updated both within and between tasks. The posterior formed after learning t tasks, serves as a prior for learning a new task. The authors suggest 3 approximation schemes, a Laplace approximation, a Hessian approximation, and a variational approximation. Finally, a set of experiments are presented comparing performance to 2 baselines, namely TOE (train of everything) and TFS (train from scratch). A particularly interesting application is to 5 standard sets of images, testing for catastrophic forgetting of previous tasks and the transferability of  information across tasks in the face of distributional shift.

Strong points: The framework is of great practical and conceptual importance. The relation to the MAML framework is clear and its incorporation within a Bayesian framework holds significant promise. The simulation results on the 5 sequential tasks provide good evidence for the effectiveness of the method.

Weak points:
The precise contribution of the work is a little difficult to discern, as it revolves very closely around previous work (MAML), Bayesian learning and Laplace approximation (e.g., Ritter et al 2018a), Variational online learning (e.g., Nguyen et al., 2018) and Hessian approximation schemes (e.g., Grosse and Martens). The approach mostly combines these methods into a single framework. It would be good to be very clear here about the novelty of the contribution.
The authors state that MAML requires all base classes to be available for sampling at each iteration. However, this does not seem to be the case for the online extension in Finn et al., 2019.
The empirical results to not compare to some of the SOTA methods in the increasingly populated domain of continual meta-learning (e.g., Finn et al., 2019), and indeed to previous methods to which the present paper is an extension (e.g., Ritter et al. 2018a).
Error bars are missing from the figures, or should, at least be mentioned.

Specific issues:
The authors use the terminology Laplace approximation for the Taylor expansion of the posterior around its maximum. As far as I am aware, the term Laplace approximation is used when this is done within an intractable Bayesian integral, in order to yield  a tractable integration. What they term a Laplace approximation is simply a Taylor expansion.
It is surprising that the BOMVI is often inferior to the simpler BOMLA (with an appropriate parameter \lambda). It would be good to discuss this issue in more detail. Can the authors suggest an approach to adapting \lambda to the data?
The title online learning is a little misleading, as the learning within tasks is done in batch. This should be compared to recently proposed fully online methods (e.g., Finn et al 2019, Khodak et al, Adaptive Gradient-Based Meta-Learning Methods, NeurIPS 2019)
The loop over i on line 4 of Algorithm 1 of the appendix has no termination point. Please clarify.

Concerns: The key concerns are incrementality WRT previous work in the field (and lack of precise elucidation of the novel aspects of the paper), lack of theory, and a lack of comparison of the empirical results to SOTA methods.

---

> ### Author Response · Authors · 2020-11-14
> **Response to Reviewer 2 (Part 1)**
>
> Thank you for your valuable feedback. We address your comments below in detail:
>
> > “The precise contribution of the work is a little difficult to discern, as it revolves very closely around previous work (MAML), Bayesian learning and Laplace approximation (e.g., Ritter et al 2018a), Variational online learning (e.g., Nguyen et al., 2018) and Hessian approximation schemes (e.g., Grosse and Martens). The approach mostly combines these methods into a single framework. It would be good to be very clear here about the novelty of the contribution.”
>
>
> Although our framework is related to Bayesian online learning (BOL) and Laplace approximation (LA) in terms of concept, we emphasize that our BOML framework is a non-trivial extension from these works. Further derivation on the Hessian approximation and re-defining the posterior objective are essential to arrive at the BOML framework. Below are our contributions in detail:
>
> + We derive a Hessian approximation in Section 3.2 and Appendix A.3.1 for the gradient-based meta-learning setting, which does not follow trivially from Martens and Grosse (2015). We further introduced the approximation required to consider the Hessian w.r.t. task-specific parameters $\tilde{\theta}$, and detailed the calculations in Appendix A.3.1.
>
> + We derive the framework in Eq.(5) by starting from the BOL principle in Eq.(16) using the support-query split and hierarchical task-parameters $\tilde{\theta}$, so that our framework is grounded based on Bayesian principles. Otherwise having only the hierarchical structure alone between meta-parameters $\theta$ and task-parameters $\tilde{\theta}$ would not be sufficient to give a strong-performing few-shot learning model that holds the knowledge learned in its posterior.
>
> > “...As far as I am aware, the term Laplace approximation is used when this is done within an intractable Bayesian integral, in order to yield a tractable integration. What they term a Laplace approximation is simply a Taylor expansion.”
>
> It is true that Laplace approximation can be used for integral approximation. We would like to emphasize that regardless of whether it is Laplace approximation for integral or posterior approximation, the underlying principle behind the derivation of LA is indeed Taylor expansion. Integral approximation performs Taylor expansion on an integrand (or rather the exponentiated integrand), whilst ours perform Taylor expansion on the log-posterior. It is reasonable to refer both of these as Laplace approximation, since they both lead to a similar outcome: integral approximation has a Gaussian integral, whereas posterior approximation has a Gaussian posterior.
>
> >”It is surprising that the BOMVI is often inferior to the simpler BOMLA (with an appropriate parameter $\lambda$). It would be good to discuss this issue in more detail.”
>
> + The main reasons for such inferiority is the variance of the Monte Carlo estimator. The final paragraph of Section 3.3 and the entire of Appendix B.2 explain the Monte Carlo estimator issue in detail and address the issue by modifying the inner loop quick adaptation.
>
> + BOMLA also performs better than BOMVI due to having better posterior approximation. Trippe and Turner (2017) compared the performances of VI with different covariance structures. In particular, they discover that VI with block-diagonal covariance performs **worse** than VI with mean-field approximate. This is because having a block-diagonal covariance prohibits variance reduction methods such as local reparameterisation trick. We will include this discussion in our paper, and will very soon update the revised paper.
>
> > “Can the authors suggest an approach to adapting $\lambda$ to the data? The title online learning is a little misleading, as the learning within tasks is done in batch.”
>
> It is true that some recent few-shot meta-learning literature considers online learning where datapoints within a task arrive sequentially for training. However, sequential tasks with batched learning within tasks is also considered as an online setting (Denevi et al., 2019) called online-within-batch setting, as mentioned in our related work Section 4. Moreover in the context of Bayesian online learning (Opper, 1998), online learning occurs on the sequential datasets $\mathcal{D}_{1}, \ldots, \mathcal{D}_T$, where each dataset corresponds to a specific task domain. The examples within each dataset are not required to arrive sequentially.
>
> > “The loop over i on line 4 of Algorithm 1 of the appendix has no termination point. Please clarify.”
>
> This loop refers to the meta-training loop where we would set the number of iterations to run for meta-training. We will change this in the paper by requiring the number of iterations J in the algorithm, and will very soon post a revision on the paper.

---

> > ### Author Response · Authors · 2020-11-14
> > **Response to Reviewer 2 (Part 2)**
> >
> > > “... The empirical results to not compare to some of the SOTA methods in the increasingly populated domain of continual meta-learning (e.g., Finn et al., 2019)...”
> >
> > Many thanks on your suggestion in adding more baseline comparisons. Before commenting on the baselines, we would like to draw your attention to the differences between our work and Finn et al. (2019). Both works may be slightly similar in the online setting, but our work is rather different in end goal and implementation method.
> >
> > + The goal of FTML (Finn et al., 2019) is to be able to decrease the number of training examples needed each round, as the tasks arrive sequentially for training. Our method focuses on being able to perform few-shot learning on unseen tasks (5-shot in sequential task Omniglot experiment and 1-shot in pentathlon experiment), whereas FTML is not designed for few-shot learning in that sense.
> >
> > + FTML follows a regret-based minimisation, whereas our framework is formulated under the i.i.d. task assumption in which different dataset domains have their own corresponding task distributions where few-shot tasks are sampled from.
> >
> > + The goal of FTML is to be able to decrease the number of training examples needed each round, as the tasks arrive sequentially for training. Our method focuses on being able to perform few-shot learning on unseen tasks (5-shot in sequential task Omniglot experiment and 1-shot in pentathlon experiment), whereas FTML is not designed for few-shot learning in that sense.
> >
> > + FTML has to keep a task buffer that accumulates data from all tasks seen so far:
> >
> >     **sequential few-shot tasks setting**
> >
> >     This may be reasonable for the sequential few-shot tasks setting if the sequence is short. For instance, the longest task sequence that Finn et al. (2019) has in the experiments is 90 tasks, whereas our Omniglot experiment has more than 250 tasks.
> >
> >     **sequential datasets setting**
> >
> >     Keeping a replay buffer would be undesirable in the sequential datasets setting, because in practice it is often not possible to store all of the data. In our pentathlon experiment, having a replay buffer would means having to keep Omniglot, CIFAR-FS, mini-Imagenet, VGG-Flowers and Aircraft as they arrive for training. In fact, the baseline comparison 'Train-on-everything' (TOE) in our experiments does indeed keep all the datasets. Keeping for 5 datasets as in our experiment is still manageable since these are relatively small academic benchmark datasets, but it would not be manageable if we have a long sequence of datasets that are as large as ImageNet. Ultimately we do not want to restrict our model in terms of the number or size of datasets it can learn from, as this would defeat the purpose of doing an online learning.
> >
> > + Explicitly keeping a memory on previous data often triggers an important question: how should the carried-forward data be processed in future task rounds, in order to accumulate knowledge? In Finn et al. (2019), few-shot tasks are sampled at each iteration from the task buffer that contains previous data. This contradicts the purpose of doing an online learning, which by definition means to update the model each round using only the new data encountered. If we have to re-train on previous data to avoid forgetting them, the training time may increase with the amount of data. We can of course clamp the amount of data at some maximal limit and perform sampling, but the final performance of such algorithm would be dependent on the samples being informative and of good quality which may vary across different seed runs. In contrast to memorising the datasets, having an implicit memory via the posterior automatically deals with the question on how to process carried-forward data and allows a better carry forward in previous experiences.
> >
> > Nonetheless, FTML will be added to our experiments as a baseline comparison, but we emphasize that our method will still be the best performing method under the few-shot learning setting, because our framework is designed for few-shot purpose (5-shot for Omniglot and 1-shot for pentathlon) whilst FTML needs about 100 datapoints or more to achieve a good performance in their rainbow MNIST experiment.
> >
> >
> > > “...and indeed to previous methods to which the present paper is an extension (e.g., Ritter et al. 2018a).”
> >
> >
> > Without our BOML framework, Ritter et al. (2018) will not be able to perform few-shot learning as we did in our paper. Ritter et al. (2018) focuses on large-scale supervised learning in contrast to few-shot learning in our paper, and is therefore not comparable to our method in terms of the setting and experiments.
> >
> >
> > > “...Error bars are missing from the figures, or should, at least be mentioned.”
> >
> >
> > We will very soon update the revised paper with the error bars included.
> >
> > ---
> >
> > We hope this clarifies the contributions of our paper and clears your doubts. We once again thank you for the useful suggestions, and hope that you would consider raising your rating score.

---

> > > ### Author Response · Authors · 2020-11-14
> > > **References**
> > >
> > > **References**
> > >
> > > G. Denevi, D. Stamos, C. Ciliberto, and M. Pontil. Online-Within-Online Meta-Learning. In Advances in Neural Information Processing Systems 32, 2019.
> > >
> > > C. Finn, A. Rajeswaran, S. Kakade, and S. Levine. Online Meta-Learning. In Proceedings of the 36th International Conference on Machine Learning, 2019.
> > >
> > > J. Martens and R. Grosse. Optimizing Neural Networks with Kronecker-Factored Approximate Curvature. In Proceedings of the 32nd International Conference on Machine Learning, 2015.
> > >
> > > M. Opper. A Bayesian Approach to Online Learning. Cambridge University Press, 1998.
> > >
> > > H. Ritter, A. Botev, and D. Barber. Online Structured Laplace Approximations for Overcoming Catastrophic Forgetting. In Advances in Neural Information Processing Systems 31, 2018.
> > >
> > > B. L. Trippe and R. E. Turner. Overpruning in Variational Bayesian Neural Networks. In Advances in Neural Information Processing Systems 30 – Advances in Approximate Bayesian Inference Workshop, 2017.

---

### Official Review · AnonReviewer1 · 2020-11-01
**Interesting problem, sound method**

**Rating:** 6
**Confidence:** 3

**Review:**

This paper presents a Bayesian meta-learning framework for sequential data. Two approximate Bayesian inference techniques, Laplace approximation and variational inference, are proposed for estimating parameters. The proposed approach helps to improve catastrophic forgetting in online settings.

The method is sound though the majority of the paper is based on other algorithms (Opper et al, Fin et al., Ngyuen et al., etc.). However, I can consider this work as the right combination of various work to solve an interesting problem.

This is framed as a typical Bayesian model selection problem. Although meta-learning is nothing more than model selection, this is apparent in section 2.2. I encourage explaining "why" performing Bayesian inference helps to mitigate the catastrophic forgetting problem well ahead in the paper.

Performing Bayesian inference over parameters, implicitly keep a memory of the history. How does the proposed approach would be different from explicitly keeping some representative samples or sufficient statistics in a memory buffer [1]? Alternatively, we can maintain an ensemble. It would perhaps be better than Laplace or mean-field approximations for this kind of complex distributions. What is the relationship of the proposed method to [2]?  Is it possible to find some meaningful online datasets such as those used in robotics or signal processing? [2] seems to have such experiments. How reasonable is the block-diagonal assumption?

[1] A. Santoro, S. Bartunov, M. Botvinick, D. Wierstra, and T. Lillicrap,“Meta-learning with memory-augmented neural networks, ICML 2016.
[2] Chelsea Finn, Aravind Rajeswaran, Sham Kakade, Sergey Levine, Online Meta-Learning, ICML 2019.

---

> ### Author Response · Authors · 2020-11-14
> **Response to Reviewer 1 (Part 1)**
>
> Thank you for your valuable feedback. We address your comments below in detail:
>
> > "I encourage explaining "why" performing Bayesian inference helps to mitigate the catastrophic forgetting problem well ahead in the paper. "
>
> We agree that this is an important point for the paper, and will very soon include an update in the paper that explains the reasons for using Bayesian inference to overcome catastrophic forgetting. Many thanks for the suggestion! Below is the explanation in detail:
>
> Bayesian online learning provides a grounded framework that naturally suggests using the previous posterior as the prior based on the recursive formula Eq.(5). An important reason to use Bayesian inference over non-Bayesian methods is to avoid training instability problems as addressed by  Antoniou et al. (2019). Bayesian online learning implicitly keeps a memory on previous knowledge via the posterior. Explicitly keeping a memory on previous data often triggers an important question: how should the carried-forward data be processed in future task rounds, in order to accumulate knowledge? In existing works such as Finn et al. (2019), few-shot tasks are sampled at each iteration from the task buffer that contains previous data. This defeats the purpose of doing an online learning, which by definition means to update the model each round using only the new data encountered. If we have to re-train on previous data to avoid forgetting them, the training time may increase with the amount of data. We can of course clamp the amount of data at some maximal limit and perform sampling, but the final performance of such algorithm would be dependent on the samples being informative and of good quality which may vary across different seed runs. In contrast to memorising the datasets, having an implicit memory via the posterior automatically deals with the question on how to process carried-forward data and allows a better carry forward in previous experiences.
>
> > "How does the proposed approach would be different from explicitly keeping some representative samples or sufficient statistics in a memory buffer [1]? "
>
> At first sight Santoro  et  al.  (2016) might look very similar to an online setting. It is important to note that the time steps in Santoro  et  al.  (2016) refers to the internal time-step within MANN, whereas our setting considers an external time-step across tasks or datasets. The goal of MANN is to train for a reader that can retrieve information quickly within task from the memory, along with the writer and the controller. When training MANN with an LSTM controller, each episode (or iteration) samples a batch of examples across the base classes, whereas our method can handle sequential tasks in meta-training. The memory mechanism in MANN is trained for quick retrieval within task, unlike our setting that goes across tasks.
>
> > "Alternatively we can maintain an ensemble. It would perhaps be better than Laplace or mean-field approximations for this kind of complex distributions."
>
> It is possible to maintain an ensemble, but the choice on how to ensemble the meta-parameters learned is rather manual. It may be reasonable to ensemble on a short dataset or task sequence, but ensemble may not do well on longer task sequence such as our Omniglot experiment that has more than 250 tasks. If we were to maintain an ensemble in this experiment, say by averaging, the meta-evaluation performance may no longer be able to build up along with the number of tasks encountered (as our method did). This is because a good choice of ensemble is very important to be able to build up on the knowledge acquired, whereas our Bayesian method is able to 'ensemble' automatically based on the framework formulation. Another way to look at our framework is that a MAP estimate objective proposes a 'direction pointer' for the meta-parameters to be able to perform well in datasets $\mathcal{D}_{1:t+1}$, whereas an ensemble requires manually setting such a 'direction pointer' by choosing for a specific ensemble method.

---

> > ### Author Response · Authors · 2020-11-14
> > **Response to Reviewer 1 (Part 2)**
> >
> > > "What is the relationship of the proposed method to [2]?"
> >
> > Our work and Finn et al. (2019) may be slightly similar in the online setting, but we are rather different in end goals and implementation methods.
> >
> > + The goal of FTML (Finn et al., 2019) is to be able to decrease the number of training examples needed each round, as the tasks arrive sequentially for training. Our method focuses on being able to perform few-shot learning on unseen tasks (5-shot in sequential task Omniglot experiment and 1-shot in pentathlon experiment), whereas FTML is not designed for few-shot learning in that sense.
> >
> > + FTML follows a regret-based minimisation, whereas our framework is formulated under the i.i.d. task assumption in which different dataset domains have their own corresponding task distributions where few-shot tasks are sampled from.
> >
> > + FTML has to keep a task buffer that accumulates data from all tasks seen so far:
> >     **sequential few-shot tasks setting:**
> >     This may be reasonable for the sequential few-shot tasks setting if the sequence is short. For instance, the longest task sequence that Finn et al. (2019) has in the experiments is 90 tasks, whereas our Omniglot experiment has more than 250 tasks.
> >     **sequential datasets setting:**
> >      Keeping a replay buffer would be undesirable in the sequential datasets setting, because in practice it is often not possible to store all of the data. In our pentathlon experiment, having a replay buffer would means having to keep Omniglot, CIFAR-FS, *mini*Imagenet, VGG-Flowers and Aircraft as they arrive for training. In fact, the baseline comparison 'Train-on-everything' (TOE) in our experiments does indeed keep all the datasets. Keeping for 5 datasets as in our experiment is still manageable since these are relatively small academic benchmark datasets, but it would not be manageable if we have a long sequence of datasets that are as large as ImageNet. Ultimately we do not want to restrict our model in terms of the number or size of datasets it can learn from, as this would defeat the purpose of doing an online learning.
> >
> > + As we mentioned in comment 1, explicitly keeping a memory on previous data often triggers an important question: how should the carried-forward data be processed in future task rounds, in order to accumulate knowledge? In Finn et al. (2019), few-shot tasks are sampled at each iteration from the task buffer that contains previous data. This contradicts the purpose of doing an online learning, which by definition means to update the model each round using only the new data encountered. If we have to re-train on previous data to avoid forgetting them, the training time may increase with the amount of data. We can of course clamp the amount of data at some maximal limit and perform sampling, but the final performance of such algorithm would be dependent on the samples being informative and of good quality which may vary across different seed runs. In contrast to memorising the datasets, having an implicit memory via the posterior automatically deals with the question on how to process carried-forward data and allows a better carry forward in previous experiences.
> >
> > Nonetheless, FTML will be added to our experiments as a baseline comparison, but we emphasize that our method will still be the best performing method under the few-shot learning setting, because our framework is designed for few-shot purpose (5-shot for Omniglot and 1-shot for pentathlon) whilst FTML needs about 100 datapoints or more to achieve a good performance in their rainbow MNIST experiment.
> >
> > > "Is it possible to find some meaningful online datasets such as those used in robotics or signal processing? [2] seems to have such experiments."
> >
> > Indeed our method can be extended to other fields of application. However further research work is needed to be able to extend the current framework Eq.(5) to other fields, as it requires modifying the log-posterior objective. In fact, this is a promising research direction where we anticipate an extension in future works!

---

> > > ### Author Response · Authors · 2020-11-14
> > > **Response to Reviewer 1 (Part 3)**
> > >
> > > > "How reasonable is the block-diagonal assumption?"
> > >
> > > The model in our paper uses the 4-layer CNN architecture proposed by Vinyals et al. (2016) that takes 4 modules with 64 filters of size $3 \times 3$, followed by a batch normalisation, a ReLU activation and a $2 \times 2$ max-pooling. A fully-connected layer is appended to the final module before getting the class probabilities with softmax.
> > >
> > > + In terms of computational cost, the most computationally intensive experiment, pentathlon, can be ran on a single 6GB RAM graphics card. In fact, the most time-consuming part of the algorithm is the meta-training routine, and not the block-diagonal approximation routine.
> > >
> > > + In terms of approximation accuracy, Martens  and  Grosse  (2015) and Ritter et al. (2018) shows a relatively large improvement of the block-diagonal approximation compared to the diagonal approximation, whilst losing only very little compared to the full Hessian.
> > >
> > >
> > > Once again we thank you for your constructive suggestions. We hope this clarifies you questions, and you would consider raising your rating score.
> > >
> > > ---
> > >
> > > **References**
> > >
> > > A. Antoniou, H. Edwards, and A. Storkey. How to Train Your MAML. In International Conference on Learning Representations, 2019.
> > >
> > > C. Finn, A. Rajeswaran, S. Kakade, and S. Levine. Online Meta-Learning. In Proceedings of the 36th International Conference on Machine Learning, 2019.
> > >
> > > J. Martens and R. Grosse.  Optimizing Neural Networks with Kronecker-Factored Approximate Curvature. In Proceedings of the 32nd International Conference on Machine Learning, 2015.
> > >
> > > H. Ritter, A. Botev, and D. Barber. Online Structured Laplace Approximations for Overcoming Catastrophic Forgetting. In Advances in Neural Information Processing Systems 31, 2018.
> > >
> > > A. Santoro, S. Bartunov, M. Botvinick, D. Wierstra, and T. Lillicrap. Meta-Learning with Memory-Augmented Neural Networks. In Proceedings of the 33rd International Conference on Machine Learning, 2016.
> > >
> > > O. Vinyals, C. Blundell, T. Lillicrap, K. Kavukcuoglu, and D. Wierstra. Matching Networks for One Shot Learning. In Advances in Neural Information Processing Systems 29, 2016.

---

### Author Response · Authors · 2020-11-16
**First Paper Revision**

We sincerely thank all the reviewers for the constructive and helpful suggestions. We have revised our paper with the following changes:

+ Added an explanation to motivate the use of Bayesian inference and why keeping an implicit memory in the posterior is superior to having an explicit task buffer in the final paragraph of Section 1 Introduction (as Reviewer 1 suggested).

+ Added a discussion on why BOMVI performs worse that BOMLA with appropriate $\lambda$ in the final paragraph of Section 6.2 experiment (as Reviewer 2 suggested).

+ Explained the i.i.d. datasets problem setup in Section 3 Overview of BOML (as Reviewer 3 suggested).

+ Added a comparison to existing problem setups (as Reviewer 3 suggested) in Section 5 Related work -- Online Meta-Learning.

+ Added references to ML-PIP/Versa, BMAML and Jerfel et al. (2019)  (as Reviewer 4 suggested) in Section 5 Related work -- Offline Meta-Learning.

+ Added a termination to Algorithms 1 and 2 (line 4) in Appendix B.1, by introducing the number of meta-learning iterations or epochs $J$ (as Reviewer 2 suggested).


The changes above are marked by "$\textcolor{red}{@@@ \text{New}}$" on the right margin of the paper. Updates for the error bars in the figures and the FTML baseline will follow in the next revision. We anticipate any further comments or suggestions on the revised paper.

---

### Author Response · Authors · 2020-11-19
**Second Paper Revision**

Following up from the first paper revision, we have further revised our paper with the changes below:

+ We've now added the FTML baseline to our experiment, as this is a major concern of many reviewers.

    + Our method outperforms the baselines. As we expected and mentioned in individual replies to [R1](https://openreview.net/forum?id=ucEXZQncukK&noteId=shopG2TZhAF), [R2](https://openreview.net/forum?id=ucEXZQncukK&noteId=HX2fMXbyFs) and [R3](https://openreview.net/forum?id=ucEXZQncukK&noteId=xVXQTFcnUQA), our methods BOMLA and BOMVI perform **better** than FTML. Since a comparison to this baseline appears to be a core concern among reviewers, we hope the reviewers now agree that our paper gives a valuable contribution and will raise their scores.

    + Since FTML is clearly under-performing in the simplest Omniglot experiment, FTML is expected to be **worse** in the more challenging pentathlon experiment. We will also include the comparison to FTML in the pentathlon experiment. Due to the time constraint, we do not anticipate having the FTML comparison in pentathlon experiment before the end of the discussion period, but we will include this in the camera-ready version.

+ Added error bands to all plots in the experiments.

+ Modified Figure 2 in the pentathlon experiment to avoid plot cluttering by showing only BOMLA with $\lambda = 100$, BOMVI, TOE and sequential MAML.

+ Moved the comparison between BOMLA of different $\lambda$ values and BOMVI to Figure 4 in Appendix C.2.


The changes above are marked by "$\textcolor{blue}{@@@ \text{New2}}$" on the right margin of the paper. Since all the reviewers' concerns have been addressed, we hope the reviewers agree that our paper provides an important contribution and is worth a higher score. We anticipate any further comments or suggestions on the revised paper.

---

### Decision · Program_Chairs · 2021-01-07
**Final Decision**

**Decision:**

Reject

**Comment:**

This paper proposes an online meta-learning algorithm. 3 out of 4 reviews were borderline. The main concern during the discussion was that it is unclear what kind of online learning this paper does. For instance, in theory, the online learner competes with the best solution in hindsight. This is a regret-minimizing point of view. The other online learning is streaming. In this case, there is no regret. The goal is a sublinear representation that is competitive with some baseline that uses all space.

After the discussion, I read the paper to understand the points raised by the reviewers. I agree that this paper is not ready to be accepted. My quick review is below:

The authors combine MAML and BOL to have online updates (not all tasks are required beforehand) and handle distribution shift. But the way of combining these is not well justified. In particular,

1) The distribution-shift story is not convincing. The reason is that the proposed algorithm is posterior-based. By definition, when you use posteriors as in (3)-(5), you assume that the datasets are sampled i.i.d. given \theta. This means independently and identically. So no distribution shift. I am familiar with Kalman filtering. For that, you need p(\theta_t | \theta_{t - 1}) in (3)-(5), which would be sufficient for tracking stochastic distribution shifts.

2) I find the use of BOL unnatural. Since MAML is gradient-based, it would be more natural to have a gradient-based online learner. Gradient descent has online guarantees and does not require i.i.d. assumptions.

3) The authors should clearly state what the objective of their online algorithm is. In particular, the informal justification of (3)-(5) as doing something similar to MAML (the paragraph around (6)) is highly confusing. I could not understand what the authors mean.

---

> ### Author Response · Authors · 2021-02-18
> **Correction reply**
>
> While we respect this as the final decision, we find some of the comments above being factually incorrect. Since the comments are publicly available in openreview, with all due respect, we find it necessary to correct the mistaken points.
>
> > 1. The distribution-shift story is not convincing. The reason is that the proposed algorithm is posterior-based. By definition, when you use posteriors as in (3)-(5), you assume that the datasets are sampled i.i.d. given $\theta$. This means independently and identically. So no distribution shift. I am familiar with Kalman filtering. For that, you need $p(\theta_t | \theta_{t - 1})$ in (3)-(5), which would be sufficient for tracking stochastic distribution shifts.
>
> The statement where "posterior by definition means i.i.d. datasets" is incorrect. Our assumption is that each dataset is independent given $\theta$, which we stated right after Eq.(5). By independence we mean $p(\mathcal{D}_1, \mathcal{D}_2 | \theta) = p(\mathcal{D}_1 | \theta) p(\mathcal{D}_2 | \theta)$. Crucially, independence does not imply that all datasets are identically generated. It seems that the Area Chair is making the assumption that independence necessarily implies that all datasets are sampled from the same underlying distribution, namely
> $\mathcal{D}_1, ...,  \mathcal{D}_t \sim p(\mathcal{D} | \theta)$. However, this is absolutely not the assumption that we are making in Eq.(3) - (5).
>
> > 2.  I find the use of BOL unnatural. Since MAML is gradient-based, it would be more natural to have a gradient-based online learner. Gradient descent has online guarantees and does not require i.i.d. assumptions.
>
> The comment where "BOL is unnatural" is rather instinct-based and not evidence-based. In fact, we've shown empirically during rebuttal that our BOL-based method outperforms gradient- or regret-based online learner (FTML).
>
> > 3. The authors should clearly state what the objective of their online algorithm is. In particular, the informal justification of (3)-(5) as doing something similar to MAML (the paragraph around (6)) is highly confusing. I could not understand what the authors mean.
>
> + The reasons for our online algorithm are in the introduction section. We've also explained the reason for using Bayesian method over regret-based method in the introduction.
>
> + Eq.(3)-(5) and paragraph around Eq.(6) are the **formal derivations** of our Bayesian online meta-learning framework, and is one of the most formal parts of our paper. The paragraph around Eq.(6) explains the choice of $p(\tilde{\theta} | \theta, \mathcal{D}_{t+1})$ and how this choice is related to MAML. This is *not* an informal justification.